# Diagnostic Approach to Enteric Disorders in Pigs

**DOI:** 10.3390/ani13030338

**Published:** 2023-01-18

**Authors:** Andrea Luppi, Giulia D’Annunzio, Camilla Torreggiani, Paolo Martelli

**Affiliations:** 1Istituto Zooprofilattico Sperimentale della Lombardia e dell’Emilia Romagna (IZSLER), 25124 Brescia, Italy; 2Department of Veterinary Science, University of Parma, 43126 Parma, Italy

**Keywords:** enteric diseases, pig, diagnosis

## Abstract

**Simple Summary:**

Pig diarrhoea is one of the most frequent health problem in modern production, which can be associated with high mortality, decreased growth rates and an increase in treatment costs. The solution for an enteric disease requires a diagnosis which is based on diagnostic criteria, that must be respected to be reliable. The veterinary practitioner has the responsibility of making a final diagnosis, and based on this to make decisions concerning the management of swine health problems. The veterinary diagnostic laboratory can be an important support providing technical assistance in performing laboratory testing and consultancy activity. The aim of this paper is to focus on the diagnostic approach of enteric disorders in pigs, from sampling to the aetiological diagnosis, taking into consideration the diagnostic criteria for the various diseases and the methods considered to be the best choice for diagnosis.

**Abstract:**

The diagnosis of enteric disorders in pigs is extremely challenging, at any age. Outbreaks of enteric disease in pigs are frequently multifactorial and multiple microorganisms can co-exist and interact. Furthermore, several pathogens, such as *Clostridium perfrigens* type A, Rotavirus and *Lawsonia intracellularis*, may be present in the gut in the absence of clinical signs. Thus, diagnosis must be based on a differential approach in order to develop a tailored control strategy, considering that treatment and control programs for enteric diseases are pathogen-specific. Correct sampling for laboratory analyses is fundamental for the diagnostic work-up of enteric disease in pigs. For example, histology is the diagnostic gold standard for several enteric disorders, and sampling must ensure the collection of representative and optimal intestinal samples. The aim of this paper is to focus on the diagnostic approach, from sampling to the aetiological diagnosis, of enteric disorders in pigs due to different pathogens during the different phases of production.

## 1. Introduction

Enteric diseases are among the most important economic problems in pig production. Neonatal and post-weaning diarrhoea represent the most frequent diseases, which can be associated with high mortality, decreased growth rates and an increase in expenses for treatment [1]. The cost of neonatal diarrhoea for herds with a mortality of 10% can be as high as EUR 134 per sow per year, and the annual cost for low-grade post-weaning diarrhoea has been estimated at EUR 40 per sow, while that for proliferative enteropathy of moderate severity has been estimated at EUR 89 per sow [2]. Multiple enteric infections during the neonatal period, as well as in the post-weaning phase, can occur concurrently, giving rise to complex clinical disease patterns and making successful control measures a challenge [3]. A correct diagnostic approach allows for differentiation between infectious and non-infectious causes of diarrhoea. It is important to remember that the majority of agents causing enteric disease in swine are part of the normal microbiota in pigs. For this reason, the detection of a potentially pathogenic agent does not correspond to a diagnosis of disease and only suggests a possible aetiology [4]. The pig intestine contains about 800 bacterial species, among which we can identify potential pathogens such as *Clostridium* spp., *Escherichia coli* and *Salmonella* spp., which are able of causing disease only under certain circumstances [5]. Interestingly, in a recent study, faecal samples from diarrhoeic and clinically healthy piglets belonging to farms suffering from neonatal diarrhoea, were tested for a panel of enteric pathogens [1]. The presence of pathogenic *E. coli*, *Clostridium perfringens* types A and C toxins (Cpα, Cpβ and Cpβ2), *Clostridioides difficile* toxins (TcdA and TcdB), Rotavirus A, B and C, porcine epidemic diarrhoea virus (PEDV) and transmissible gastroenteritis virus (TGEV), was investigated using molecular methods and bacteriology. Rotavirus type A was the only agent that could be statistically correlated with diarrhoea. Significant differences for the other viruses and bacteria analysed between the diseased pigs and the apparently healthy pen-mates were not found.

The diagnostic approach must therefore take into account the pathogen’s biology and ecology, the anatomic location from which the pathogen is detected, and the available and appropriate diagnostic tools to demonstrate a correlation between a pathogen and a diseased status [4].

The aim of this paper is to focus on the diagnostic approach, from sampling to aetiological diagnosis, of enteric disorders in pigs caused by the most important pathogens (bacteria, viruses and parasites) causing diseases during the different phases of production.

## 2. The Diagnostic Approach to Enteric Diseases

### 2.1. General Considerations

The diagnostic pathway requires a systematic collection of information regarding the clinical history, the evaluation of clinical signs and gross lesions, sample collection, and the choice of laboratory tests for one or more suspected diseases (Figure 1).

The clinical history will guide the veterinarian through the diagnostic process. Enteric diseases of infectious origin have an age-related distribution that reflects the epidemiology of each pathogen on the farm [6]. For this reason, the age of the animals affected in an outbreak can help the veterinarian to better focus and investigate the most probable aetiological agents, based on the age at which they are most commonly found (Figure 2).

### 2.2. Clinical Examination of Enteric Diseases in Pigs

The most important clinical manifestation of enteric disease in pigs is diarrhoea. Changes in colour (yellow, grey, bloody, etc.) and consistency (watery, creamy, etc.) can help the clinician to establish a differential diagnosis. Vomiting is another important clinical sign that can be observed in enteric disease and is usually associated with infections sustained by enteric viruses [6].

The number of affected litters, the number of affected piglets per litter and the parity of the sows whose litters are affected can also help to understand the origin of the problem. As a general rule, enteric disease due to endemic aetiological agents tends to affect mainly litters belonging to first-parity sows, while the introduction of a new agent results in a generalized outbreak [6]. 

### 2.3. Necropsy and Gross Evaluation

Following clinical evaluation, the most effective approach is to select three to five appropriate pigs (acutely affected and with representative clinical signs) that have not received antimicrobial treatments. Selected pigs should be humanely euthanized and submitted to post-mortem examination, through necropsy, to perform gross morphological diagnosis of the lesions and to collect samples for further diagnostic investigation (small intestine, colon, ileocecal valve and mesenteric lymph nodes). The necropsy can provide useful information for the orientation of the diagnosis, even if in most cases it is not possible to reach a definitive diagnosis based solely on the macroscopic findings observed. The information collected during the necropsy, such as the type of enteritis (catarrhal, fibrinous, necrotic, etc.), the localisation of the lesions (small and/or large intestine) (Figure 3) and its distribution (focal, diffused, segmental, etc.) can improve the formulation of a differential diagnosis [6].

A complete set of diagnostic tests is recommended, including bacteriology, PCR and histopathology. Taking shortcuts can result in missing the real cause of the problem [7].

### 2.4. Sampling

Correct sampling represents a key moment in the diagnostic process and guarantees reliable results from subsequent laboratory investigations. The choice of the animals to be sampled, the number of samples and the type of samples to be taken, in relation to the clinical and gross lesions observed, are the key points of a correct sampling. 

After collection, samples must be stored correctly. Improperly stored samples can invalidate the results of diagnostic tests. The diagnostic pathway of enteric disease includes sampling from dead and/or live animals (Figure 4).

Fresh samples of unopened segments of the small intestine (in particular the ileum and jejunum) and large intestine should be taken from dead and necropsied animals, kept in a different container or bag away from other tissues to avoid contamination and sent to the laboratory for bacteriology and PCR. Fresh samples should be stored at +4 °C and should arrive at the laboratory in less than 24 h. Tissues should be collected following standardized procedures (Table 1) and fixed in 10% buffered formalin for histology [4]. Since autolysis of the gut after death occurs quickly, tissues obtained from dead animals are usually not suitable for histopathological analyses [6]. For this reason, tissues from euthanized animals are preferred to animals found dead for laboratory submissions. So, the post mortem examination should be performed in animals found dead, but sample collection should preferably come from euthanized animals.

### 2.5. Diagnostic Investigations

The diagnosis of enteric disorders in pigs requires the combination of several diagnostic techniques. Some of them, such as quantitative bacteriology and quantitative real time PCR, can demonstrate the quantitatively significant presence of an aetiological agent, necessary for a definitive diagnosis. In many cases the sole isolation of the pathogen cannot allow a conclusive diagnosis. Confirmation of pathogenicity by the demonstration of virulence factors, such as genes encoding for toxins, is needed (e.g., enterotoxigenic *Escherichia coli*, *C. perfrigens*, *C. difficile*, etc.). The use of histology is of fundamental value to correlate typical microscopic lesions with the isolation of a certain aetiological agent, and to verify the co-localization of specific antigens (belonging to potential aetiological agents) associated with lesions using immunohistochemistry techniques.

Serology is of little use in the diagnosis of most, if not all, enteric diseases of pigs. This is due in many cases to the lack of available serological tests available. Furthermore, the acute presentation of most enteric disease outbreaks is not compatible with a serological diagnosis based on paired samples collected in acutely affected pigs and 3 weeks after in convalescent animals [6].

## 3. Diagnostic Approach to the Main Enteric Diseases of Pigs

### 3.1. Enteric Colibacillosis

#### 3.1.1. Aetiology and Clinical Presentation

Neonatal colibacillosis is caused by enterotoxigenic *E. coli* (ETEC) possessing surface proteins called fimbriae, identified as F4 (k88), F5 (k99), F6 (987P) and F41. The fimbriae allow the microorganism to adhere to specific receptors on the brush borders of the small intestine’s enterocytes [8]. Susceptibility to ETEC F5, F6 and F41 decreases with age due to a reduction in the number of active receptors located on the intestinal epithelial cells as animals age. Most ETEC strains of neonatal colibacillosis produce the heat stable enterotoxin STa that is responsible for secretory diarrhoea, leading to electrolyte and fluid secretion [9].

Neonatal colibacillosis is observed most commonly in piglets aged from 0 to 4 days of life, and generally when endemic, litters from first-parity sows could be more involved due to a lack of protection by passive immunity [10]. Mortality is greatest in pigs less than 4 days old, while in pigs older than 7 days morbidity and lethality are much lower [11]. Morbidity on average is 30–40%, but may be as high as 80% in some herds, while lethality can reach 70% in affected litters.

Diarrhoea may be very mild with no evidence of dehydration or may be profuse and is characterized by [9]:(1)alkaline pH;(2)watery to creamy consistency;(3)a distinctive smell;(4)white to yellow colour;(5)possible various shades of brown.

Affected pigs are usually depressed with a reduced appetite, and may vomit or show a rough, sticky, wet hair coat. Death occurs commonly 12–24 h after the onset of diarrhoea [11]. In a low proportion of pigs, death occurs before the development of diarrhoea. In severe cases, dehydration can determine the loss of 30–40% of total body weight and piglets can show sunken eyes and the exaggeration of bony prominences. The anus and perineum may present redness due to the contact with the alkaline diarrhoeic faecal material (Figure 5).

Pigs with less severe dehydration may continue to drink and, if treated appropriately, recover with only minimal long-term effects [8].

Enterotoxigenic *E. coli* strains responsible for post-weaning diarrhoea (PWD) possess fimbriae F4 and F18, with some rare exceptions, and produce one or more of the following known enterotoxins: heat-stable enterotoxins STa, STb, the heat-labile enterotoxin LT and the enteroaggregative *E. coli* heat-stable enterotoxin (EAST1) [8].

PWD due to ETEC is commonly observed 2–3 weeks after weaning, even if cases recorded 6–8 weeks after weaning are not an exception. Clinical signs are characterized by yellowish, grey or slightly pink watery diarrhoea with a characteristic smell, generally lasting one week. Affected pigs are usually depressed with a reduced appetite and a rough, sticky, wet hair coat [8]. Sudden death can occur, particularly at the start of the outbreak, and dead pigs are usually dehydrated with sunken eyes; mortality can reach up to 25% [10]. In addition to ETEC strains, more rarely, enteropathogenic *E. coli* (EPEC) can be isolated from cases of PWD.

#### 3.1.2. Pathological Changes

Colibacillosis cannot be readily differentiated from the other common causes of diarrhoea based solely on gross findings, without laboratory investigations [12]. Piglets affected by neonatal and post-weaning colibacillosis usually show a dilated stomach full of clotted milk or dried feed, with hyperaemia of the fundus and venous infarcts on the greater curvature (Figure 6).

The small intestine is usually dilated, slightly oedematous and hyperaemic with a diarrhoeic content with a characteristic smell (Figure 7 and Figure 8). The mesenteric lymph nodes are enlarged and frequently hyperaemic. These lesions, even if not pathognomonic, are suggestive of enteric colibacillosis and help the pathologist in the choice of subsequent laboratory tests [9].

The microscopic lesions in pigs infected by ETEC consist of multifocal aggregates of rod-shaped, basophilic bacteria on the brush borders of the small intestine enterocytes (Figure 9). Mild villous atrophy and an increased number of neutrophils may be observed in the superficial lamina propria [9]. 

Histology shows multifocal colonisation of the brush border of mature enterocytes by *E. coli* arranged in palisades with enterocyte degeneration and light to moderate inflammation of the lamina propria in pigs infected with EPEC [9].

#### 3.1.3. Diagnostic Tools and Criteria

The diagnosis of enteric colibacillosis is based on bacteriological examination of samples of luminal content (first choice), faeces or rectal swabs. The isolation of the pathogenic *E. coli* strain involved in the outbreak by bacteriology, its quantification (pure culture or not), the identification of virulence factors (usually by PCR), and histopathology as a complementary analysis, represent the steps to take for diagnosis [10].

Faeces, rectal swabs or intestinal contents are inoculated onto blood agar, where the quantitatively significant presence of haemolytic colonies can be used for a presumptive diagnosis of ETEC diarrhoea [10], and on media which are selective for *Enterobatteriaceae* such as McConckey agar or Hektoen agar. These selective media allow differentiation of lactose fermenting (such as *E. coli*) from lactose non-fermenting Gram-negative enteric bacilli.

In general terms, ETEC isolated from cases of neonatal colibacillosis can appear as haemolytic (ETEC F4) or non-haemolytic (ETEC F5, F6 and F41) colonies on blood agar plates, while ETEC isolated from cases of PWD are mostly haemolytic (ETEC F4 or F18) [8]. EPEC strains are always non-haemolytic [9].

The detection of pathogenic strains does not justify the disease in every case and it is important to consider that pathogenic *E. coli* can also be isolated from the gut habitat of healthy hosts [13]. For this reason, the quantification of the pathogenic *E. coli* isolated in high concentration, in pure or nearly pure culture, from the small intestine (ileum and jejunum) is indicative of enteric colibacillosis [10]. The identification of virulence genes encoding for the fimbriae and toxins of the isolated strain by polymerase chain reaction (PCR) is crucial to ascertain its role in the clinical disorder observed (Figure 10).

Primers recognising genes encoding for toxins (STa, STb, LT and EAST1) and fimbriae (F4, F5, F6, F18 and F41) of ETEC, for the outer membrane protein Eae or intimin in enteropathogenic *E. coli* (EPEC) and for Stx2e toxin in EDEC (*E. coli* involved in oedema disease) strains, are available and can be used to perform PCR assays for daily routine diagnostics [14].

### 3.2. Clostridiosis

#### 3.2.1. *Clostridium perfringens* Type C

##### Aetiology and Clinical Presentation

Infection with *Clostridium perfringens* type C occurs worldwide, causing fatal necro-haemorrhagic enteritis mostly in neonates, even if cases of clostridiosis can be observed until 3 weeks of age [15,16]. The disease can spread rapidly in a herd, and mortality in affected piglets from non-vaccinated herds can reach 100% [16]. Clinical signs may be peracute, acute or chronic. The evolution of the disease is influenced by the:(1)piglet’s immune status;(2)age of affected piglets (peracute and acute disease affect piglets mainly within the first 3 days of life);(3)virulence of the *C. perfringens* strain involved.

Peracutely affected piglets can develop haemorrhagic diarrhoea that begins 8–22 h after exposure to *C. perfringens* type C [17], or can show depression and rapid onset of death [16]. Piglets are weak, reluctant to move, hypothermic (rectal temperature falls to 35 °C or below) and with abdominal skin that may darken before death [15]. 

Acutely affected piglets showing haemorrhagic diarrhoea may die soon after the onset of clinical signs or 24–48 h after the onset of symptoms [15,16]. They have brown–red diarrhoea containing grey shreds of necrotic material and show dehydration. Perineal cutaneous lesions may occur and piglets rapidly show weakness and die [17]. 

Sub-acutely and chronically affected pigs show non-haemorrhagic diarrhoea (faeces are yellow to grey and mucoid), reduced growth and emaciation. These piglets usually die after several weeks or must be euthanized due to unthriftiness [15,16].

##### Pathological Changes

Lesions usually have a segmental distribution involving the small intestines, sometimes including the spiral colon. The latter may be the only intestinal tract affected.

In peracute and acute cases, the lesions in the small intestine include segmental mucosal necrosis, haemorrhages, emphysema and mucosal fibrino-necrotic exudate [15,16]. In most cases, lesions are confined to the jejunum. However, they can extend into the ileum [16]. Deposition of urate crystals in the kidneys is common [15].

Histologically, peracute disease is characterised by the haemorrhagic necrosis of the intestinal wall [17]. Large numbers of Gram-positive rods can be observed covering the outlines of necrotic villi. Thrombosis and necrosis of small vessels in the lamina propria and submucosa are consistently present [16]. Systemic capillary thrombosis, though uncommon, and secondary to terminal disseminated intravascular coagulation, may be observed in different tissues such as the lungs, liver, spleen and kidney [12].

Mucosal fibrinous pseudomembranes with a marked intestinal inflammatory response after the initial mucosal necrosis [15,16] and marked necrosis of the mucosa, which can extend to transmural intestinal necrosis with fibrinous peritonitis [16], are the main gross lesions in chronic disease. Histologically, the necrotic mucosa is markedly infiltrated by neutrophils and mixed populations of Gram-positive and Gram-negative bacteria, most likely representing overgrowth of commensal flora, are visible histologically in subacute to chronic cases [16].

##### Diagnostic Tools and Criteria

A presumptive diagnosis of type C clostridiosis in neonatal piglets can be based on [17]:(1)presence of haemorrhagic diarrhoea;(2)rapid death;(3)segmental necro-haemorrhagic or fibrino-necrotic enteritis.

Quantitative bacteriology performed on intestinal contents and/or faeces, in order to demonstrate large numbers of *C. perfringens* type C, is recommended to confirm a final diagnosis of clostridiosis [16]. On blood agar, after 24 h of anaerobic incubation at 37 °C, *C. perfringens* type C, as all other types of *C. perfringens*, normally forms a characteristic double zone of haemolysis. Following culture, the genotyping of isolates using a multiplex PCR test to detect genes for the major toxins is the most common method to identify *C. perfringens* type C (Figure 11).

Histopathology can further confirm the diagnosis. It is important to highlight that animals that have died spontaneously show a certain degree of autolysis that can mask the necrotizing histologic lesions in the mucosa, depending on the post-mortem interval until necropsy. This autolysis can destroy the architecture of the tissue and antigens present in it, thus immunohistochemistry to detect *C. perfringens* beta-toxin is not the method of choice for routine testing [16].

Commercial enzyme-linked immunosorbent assay (ELISA) kits for the detection of CPB in intestinal contents are available. The sensitivity of the toxin to trypsin, pepsin, and possibly other proteases, can led to false-negative results, in particular if the samples are not fresh [16].

#### 3.2.2. *Clostridium perfringens* Type A

##### Aetiology and Clinical Presentation

*C. perfringens* type A is a member of the normal microbiota of the swine intestine, and even if there is very little information available about the role of *C. perfringens* type A in diarrhoeal diseases of pigs [15], it is considered a cause of enteric disease in neonatal and, occasionally, in weaned pigs [17]. Conditions leading to clinical manifestation of infection are the presence of a large number of *C. perfringens* type A, reaching 10^8^ to 10^9^ CFU/g of intestinal contents and toxin production. *C. perfringens* type A produce CPA toxin, and some strains also produce CPB2 toxin [17]. Enteric disease caused by *C. perfringens* type A in pigs has been associated with non-haemorrhagic mucoid diarrhoea in suckling piglets [17].

##### Pathological Changes

No consistent lesions have been reported in *C. perfringens* type A-associated diarrhoeal disease in neonatal pigs. The lack of gross and histological lesions in *C. perfringens* type A enteritis makes definitive diagnosis difficult. Uzal and Songer (2019) described the normal length and morphology of the intestinal villi, infiltrates of few neutrophilic aggregates in the apical lamina propria and large numbers of bacilli (*C. perfringens*-like) in the lumen [15]. Songer and Uzal (2005) reported necrosis of the small intestine mucosa with villous atrophy, the presence of fibrino-necrotic pseudomembranes and occasional serositis in a low proportion of affected pigs [17]. The small intestine can contain chyle, and colonic mesentery can be slightly oedematous [18].

##### Diagnostic Tools and Criteria

*C. perfringens* type A is part of normal intestinal microbiota, and for this reason can be cultured from diarrhoeic and asymptomatic neonatal pigs [19]. Currently, no criteria are available to establish a definitive diagnosis of clostridiosis in pigs due to *C. perfringens* type A. Some criteria, if satisfied, are considered suggestive of clostridiosis type A (Figure 12) [15]:(1)non-haemorrhagic diarrhoea of otherwise unexplained origin;(2)isolation of large numbers of *C. perfringens* type A from the small intestine using quantitative bacteriology;(3)detection of genes for toxins.

**Figure 12 animals-13-00338-f012:**
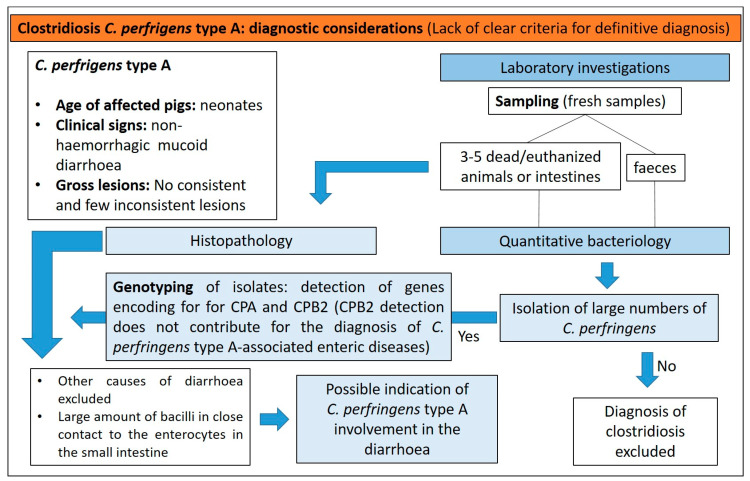
Diagnostic algorithm for the diagnosis of clostridiosis due to *C. perfringens* type A in pigs.

#### 3.2.3. *Clostridioides difficile*

##### Aetiology and Clinical Presentation

*Clostridioides difficile*-associated disease (CDAD) affects numerous animal species, including pigs. In pigs, *C. difficile* causes disease mostly between 1 and 7 days of age, usually presenting a history of early-diarrhoea, mild abdominal distension, scrotal oedema or sudden death. Some of the affected animals can have normal faeces, or they can even be constipated [20]. In less severely affected animals, diarrhoea, abdominal distension, decreased appetite and poor growth may be evident [20]. A subclinical disease in some animals has been suggested [21].

##### Pathological Changes

The most common and characteristic lesion is the oedema of the mesocolon (Figure 13). The cecum and colon usually show diarrhoeic or pasty yellow to brown contents, while the mucosa of these segments may have multifocal-to-coalescing erosions or ulcers with fibrino-necrotic exudation. Less commonly, scrotal oedema, ascites, hydropericardium and hydrothorax can be observed.

Microscopically, suppuration in colonic lamina propria, colonic serosal and mesenteric oedema and infiltration of mononuclear inflammatory cells and neutrophils, as well as segmental erosion of the colonic mucosal epithelium and “volcano” lesions (exudation of neutrophils and fibrin into the lumen), are highly suggestive of CDAD in piglets [12,17]. Occasionally, deeper necrosis of the mucosa and colonic wall may occur [12].

##### Diagnostic Tools and Criteria

*C. difficile* in the intestinal tract of healthy piglets shows a high prevalence. For this reason, a culture of *C. difficile* and the detection of toxins alone are of little diagnostic significance unless associated with compatible clinical signs and gross and histopathological findings. The diagnosis of CDAD in pigs should be established on (Figure 14) [15,22]:(1)compatible clinical history;(2)clinical signs;(3)gross and microscopic findings;(4)detection of TcdA and/or TcdB in faeces or colonic contents;(5)isolation by bacteriology and detection by PCR of genes encoding for TcdA and/or TcdB toxins is important for confirmation, considering that some isolates produce only TcdB or no toxins at all.

**Figure 14 animals-13-00338-f014:**
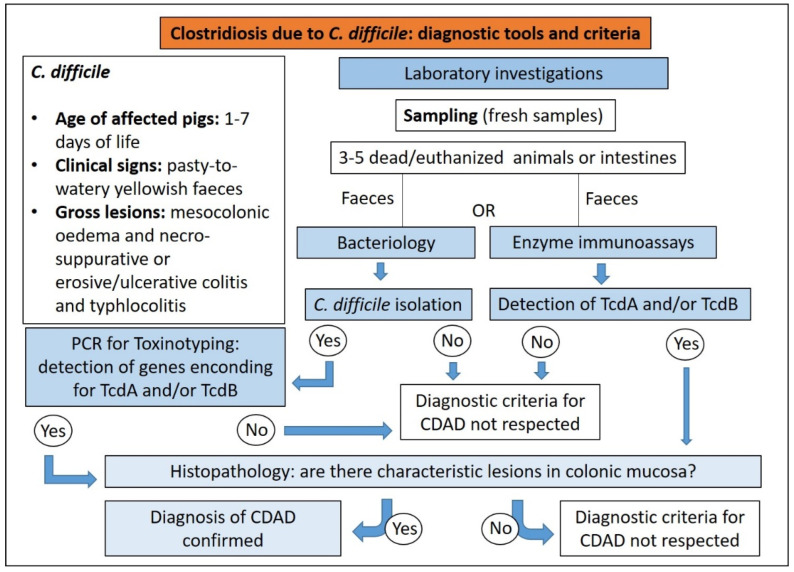
Diagnostic algorithm for the diagnosis of clostridiosis due to *C. difficile* in pigs.

Histopathology is a very useful ancillary technique for post-mortem diagnosis of CDAD, with the characteristic “volcano lesion” being suggestive of a possible involvement of *C. difficile* [22].

Recently, enzyme immunoassay for the detection of glutamate dehydrogenase (GDH) in diarrhoeic faeces is a highly sensitive method to demonstrate *C. difficile* infection in piglets [23]; even if it does not differentiate between toxigenic and non-toxigenic strains [24], it could be useful as a screening method for piglets [22].

### 3.3. Intestinal Enterococcosis

#### 3.3.1. Aetiology and Clinical Presentation

Enterococci are part of the normal intestinal microbiota, but some strains, showing the capacity to colonise the apical surface of the enterocytes of the small intestine of young animals, have been associated with diarrhoea in different animal species [25]. Watery to creamy diarrhoea caused by *Enterococcus durans* [26] and by *Enterococcus hirae* [27] have been reported in piglets between 2 and 20 days of age [26,27]. These organisms can adhere to the surface of villous enterocytes in the small intestine, but the pathogenesis of diarrhoea is unclear [26,27].

In a comparative study, Larsson et al. (2014) demonstrated that the intestinal colonisation by *E. hirae*, accompanied by mucosal lesions, was observed in diarrhoeic pigs only [27]. Kongsted et al. (2018) reported that the detection of *E. hirae* was not associated with neonatal piglet diarrhoea; however, the results of the study suggested that massive overgrowth could be part of the pathogenesis in some cases of neonatal diarrhoea [28].

#### 3.3.2. Pathological Changes

Gross lesions in diarrhoeic neonatal piglets, associated with entero-adherent *E. hirae*, are characterised by watery to creamy contents in the distal colon, dilated large intestine and congestion of the small intestine’s blood vessels [27].

Histopathological changes described by Larsson and colleagues (2014) in diarrhoeic piglets with diarrhoea associated with *E. hirae* infection were characterised by the small intestinal colonisation by Gram-positive entero-adherent cocci, villous atrophy and mild epithelial lesions, including increased apoptosis of enterocytes [27].

Other microscopic findings were [27]:(1)detachment of enterocytes with a rounded shape and eosinophilic cytoplasm;(2)foci of shallow mucosal erosions with exudation of fibrin and neutrophils;(3)capillary microthrombi in the lamina propria of villous tips;(4)altered villous:crypt ratio in the proximal or distal jejunum.

*E. villorum* has been identified in piglets with diarrhoea, but without association with macroscopic and microscopic findings [25].

#### 3.3.3. Diagnostic Tools and Criteria

Neonatal diarrhoea associated with isolation in blood agar of a pure or nearly pure culture of Gram-positive bacteria can allow a presumptive diagnosis of diarrhoea due to *Enterococcus* spp. The isolation must be followed by the identification as *E. hirae*, *E. durans* and *E. villosum* using biochemical systems, PCR or Matrix Assisted Laser Desorption/Ionisation Time-Of-Flight Mass Spectrometry (MALDI-TOF MS) [27]. Other confirmatory diagnostic tools are: 16S rRNA gene analysis, fluorescence in situ hybridisation (FISH) and immunohistochemistry. The histological evaluation of the jejunum and ileum from affected piglets can be an important confirmatory diagnostic tool, revealing numerous Gram-positive cocci adhered to the villous epithelial cells [29], together with other possible lesions as described above.

### 3.4. Coccidiosis

#### 3.4.1. Aetiology and Clinical Presentation

*Cystoisospora suis* is the most important aetiological agent of coccidiosis in pigs, usually affecting pigs in the second week of life [30].

Faeces of diseased piglets are initially pasty, yellowish-grey in colour and become more fluid as the infection progresses [31]. The faeces are initially loose or pasty. Enteritis leads to malabsorption followed by a reduction of weight gain or, in severe cases, emaciation. Reduced weight gain can also be observed in piglets without diarrhoea [32]. Litters within the farrowing house show variable severity of clinical signs. In fact, symptomatic and asymptomatic piglets can be observed within the same litter. Morbidity is normally high, but mortality can range from low to moderate.

Typically, individual animals show a biphasic excretion pattern upon infection with a steep onset at the beginning of patency, usually five to six days after infection [33].

#### 3.4.2. Pathological Changes

The severity of the pathological changes is correlated with the infection dose and the age of the piglets, as younger animals are more affected than older ones [31]. Macroscopic changes include non-haemorrhagic enteritis involving the jejunum and ileum, with oedematous mucosa, occasionally with fibrino-necrotic membranes attached to it and, in high-dose infections, haemorrhagic enteritis [31]. Microscopic lesions consist of villous atrophy, villous fusion, crypt hyperplasia and necrotic enteritis [34].

#### 3.4.3. Diagnostic Tools and Criteria

Diarrhoea in piglets 7–14 days old that does not respond to antimicrobial treatments is suggestive of coccidiosis [31]. In the differential diagnosis the following should be considered:(1)Enterotoxigenic *E. coli*;(2)Coronaviruses;(3)Rotavirus;(4)*C. perfringens* type C;(5)*Strongyloides ransomi*.

Liquid faecal samples are likely to contain less oocysts than pasty faecal samples [34]; in fact, the correlation between faecal consistency and oocyst excretion is weak. For this reason, is not advisable to primarily take samples from piglets with diarrhoea, as such samples may contain few oocysts [33].

Oocyst identification has traditionally been carried out with faecal flotation. However, as faecal fat may make the identification difficult [33], a different flotation medium should be used to increase the sensitivity. Detection of oocysts in smears under light microscopy is of poor sensitivity and specificity [35]. The most common flotation medium for *C. suis* oocysts is Sheather’s sugar solution or modifications of it. An alternative flotation medium (500 g of glucose in 1000 mL saturated sodium chloride solution) has been recommended [36,37,38]. The use of Carbol-fuchsin staining or other alternative methods such as Auramine O, Löffler’s methylene blue, Lugol’s solution, May-Grünwald, Ziehl-Neelsen or Gentiana violet, can aid the detection of oocysts in faecal smears [33]. For the quantification of oocysts in faecal material, enumeration in a McMaster chamber is the standard [39].

Oocyst detection using faecal smears requires samples taken repeatedly (sampling three times at 7, 14 and 21 days of age or at 10, 15 and 20 days of age), in order to determine the extent and intensity of infection on a farm [33]. Faecal smears used for autofluorescence microscopy or stained with different methods demonstrated, for the autofluorescence method, the lowest detection limit [33]. Molecular tools have been used to detect and differentiate stages in faeces with high sensitivity and specificity by using PCR, nested PCR and real-time quantitative PCR (qPCR) assays [40,41]. Histologic diagnosis of *C. suis* in tissue sections should be used to confirm the diagnosis [6]. The essential diagnostic tools and criteria for the diagnosis of coccidiosis are reported in Figure 15.

### 3.5. Rotavirosis

#### 3.5.1. Aetiology and Clinical Presentation

Rotaviruses (RVs) are an important cause of diarrhoea in pigs. Ten rotavirus (RV) groups, from A to J, have been described and five of them (RVA, RVB, RVC, RVE and RVH) have been reported in pigs [42]. The most important group in swine is RVA, showing the higher prevalence and pathogenicity. RVC has been described as an important cause of diarrhoea in neonatal pigs [43], while RVB was identified as a cause of infection in older animals [42]. The presence of porcine RVH has been confirmed as a cause of diarrhoea in pigs in Japan, Brazil and the US [42].

Field outbreaks of RV diarrhoea can be observed in neonatal pigs, but are found more commonly in 2 to 6 weeks old pigs, as most sows provide degrees of colostral protection [44].

The outcome of the RV infection (ranging from subclinical to severe disease) can be influenced by the [45,46]:(1)RV strain;(2)age of pigs;(3)immune status;(4)overall herd health;(5)presence of secondary bacterial or viral infections.

Morbidity may be 20%, with mortality as high as 15% [44]. Clinical signs include profuse yellow–white watery diarrhoea, with undigested milk, lethargy, vomiting and anorexia accompanied by poor average daily gain and rapid weight loss [46,47]. Diarrhoea in piglets (7 days old or younger) can persist for 1–10 days, while in older pigs the disease is generally less severe and shorter in duration [46].

#### 3.5.2. Pathological Changes

Lesions due to Rotavirus types A, B and C are similar and can be segmental, involving only short sections of intestine. The stomach can be swollen, containing undigested milk [46], while the small and large intestines show a thinning of the intestinal wall due to villous atrophy and an excessive dilatation, with the accumulation of watery, yellow or grey diarrhoeic faeces in the lumen [11].

The microscopic lesions observed in infected swine at the jejunal and ileal level include a reduction in the villous-to-crypt length ratio, villous loss and fusion, villous atrophy, epithelial vacuolation, cuboidal-to-flat epithelial cells at the tips of villi and epithelial desquamation in the small intestine [48]. Shortening of villi is accompanied by elongated intestinal crypts with cellular hyperplasia [46].

#### 3.5.3. Diagnostic Tools and Criteria

Clinical signs caused by Rotavirus cannot be differentiated clinically from those caused by other enteric pathogens, and, therefore, the diagnosis requires laboratory testing [46]. The diagnosis of rotavirosis should consider that Rotavirus can be detected in both diarrhoeic and, in cases of subclinical infection, in asymptomatic pigs [46,49].

Reverse transcriptase-polymerase chain reaction (RT-PCR) or reverse-transcriptase quantitative polymerase chain reaction (RT-qPCR), to detect and distinguish between RVA, RVB and RVC from faecal or stomach content, are commonly used [50].

The implementation of multiplex RT-PCR tests can allow the simultaneous detection of several enteric viruses in clinical samples [51]. Multiplex real-time RT-PCR (rRT-PCR) methods can simultaneously detect RVA, RVB and RVC [52] and correctly assess the real prevalence in case of neonatal diarrhoea outbreaks. As an example, the low prevalence reported worldwide for RVB may not be a consequence of a low infection rate, but rather due to diagnostic gaps [53].

Several commercial ELISA antigen kits are available. These tests are fast and reliable screening tools to test multiple faecal samples. Unfortunately, not all porcine RV species have commercially available ELISAs and most of them are only specific for the RVA species [50].

Intestinal tissues can be evaluated using histopathology to detect characteristic lesions, and the use of immunohistochemistry allows visualisation of virus particles within cells. There are commercially available antibodies to RVA, RVB and RVC [50].

In situ hybridisation RNA-based chromogenic technique (ISH-RNA) has been validated and used to detect and subtype RVA, RVB and RVC [54]. Diagnostic criteria for rotavirosis diagnosis are reported in Figure 16.

### 3.6. Porcine Enteric Coronaviruses (PECs)

#### 3.6.1. Aetiology and Clinical Presentation

Porcine enteric coronaviruses (PECs), including transmissible gastroenteritis virus (TGEV), porcine epidemic diarrhoea virus (PEDV), porcine deltacoronavirus (PDCoV) and swine acute diarrhoea syndrome coronavirus (SADS-CoV) are recognised as causes of gastrointestinal disease in pigs [55].

PEDV, PDCoV and SADS-CoV are newly emerging or re-emerging PECs representing a serious problem in pig production [56]. Neonatal piglets show the highest susceptibility to PECs due to [57]:(1)less acidic pH in the stomach compared to older pigs;(2)renewal of enterocytes lining the intestinal villi from progenitor cells in the intestinal crypts is less rapid than in older pigs;(3)the neonatal immune system is not fully mature;(4)higher vulnerability to the electrolyte and fluid imbalances that result from poor digestion and severe malabsorption diarrhoea.

TGE is a highly contagious viral disease of swine characterised by gastrointestinal clinical signs, such as vomiting and severe diarrhoea. The severity of the disease is age dependent and in piglets less than 2 weeks of age the lethality can reach 100%. The clinical impact of TGE was lessened by the appearance and diffusion of porcine respiratory coronavirus (PRCV), a deletion mutant of TGEV [58]. For example, in the USA the percentage of TGEV positive cases was 4.0% in 2008, increased to 6.8% in 2010, and decreased to 0.4% in 2014 [59].

Disease caused by PEDV is very similar to TGE and peaks in the late fall and early winter months in cold climates, then declines in late spring, summer and early fall [60]. On breeding farms, pigs of all ages become ill. Infection of neonatal pigs usually results in significant mortality rates (approaching 100%) caused by malabsorptive diarrhoea and dehydration as a result of small intestinal enterocyte necrosis and villous atrophy.

Infection of grow-finish animals results in moderate morbidity but typically little to no mortality, with vomiting and mild to moderate diarrhoea, watery faeces, and often severe anorexia and depression. Older pigs recover after about 1 week. Sows and gilts may have mild diarrhoea, reduced activity and poor appetite upon primary exposure. Sows and fattening pigs usually show diarrhoeic watery faeces, depression and anorexia [58,60].

The acute phase of the disease is self-limiting on a breeding farm and generally lasts 3–4 weeks (but may be much longer in large breeding farms with multiple separated units). After this interval piglets are protected by the lactogenic immunity developed by the pregnant sows. After the resolution of the outbreak, diarrhoea may be observed recurrently in weaned pigs [58]. Interestingly, the virus can be shed with faeces prior to the presentation of clinical disease and after the resolution of clinical signs [60].

PDCoV can infect pigs of different ages and, as for PEDV, piglets are more susceptible [61]. Clinical signs of PDCoV infection can include diarrhoea, dehydration, variable vomiting and mortality in neonatal piglets. These clinical manifestations are similar to other swine enteric pathogens such as PEDV and TGEV [62]. The clinical impact, prevalence and disease severity of PDCoV in the field are milder than those of PEDV [63].

SADS-CoV is considered to be the causative agent of fatal swine acute diarrhoea syndrome (SADS) with clinical manifestations of severe, acute diarrhoea and rapid weight loss in piglets [64]. Mortality has been reported as 90–100% in pigs up to 5 days old, decreasing to 5% in pigs above 8 days of age [56].

#### 3.6.2. Pathological Changes

Necropsies of TGEV naturally infected pigs show extreme thinning of the small intestine [65] (Figure 17), due to mild–moderate to severe villous atrophy and hyperplasia of crypts in the jejunum and ileum [66].

Lesions produced by PEDV and TGEV infections are very similar and are mainly distributed amongst duodenum and proximal and mid-jejunum, that appear distended with watery and yellow diarrhoea [67,68]. Microscopically, the exfoliation of superficial enterocytes, karyomegaly, vacuolation, the formation of syncytia, necrosis and atrophy have been described [58,67]. The villous height to crypt depth ratio in the small intestine is decreased, while the large intestine shows no pathological changes [58,68].

PDCoV lesions are similar than those caused by PEDV, but milder [69]. After experimental infection, the small intestine, cecum and colon appear thin-walled and gas-filled, with a yellow, soft to watery content, while the stomach usually contains curdled milk [70,71]. The jejunum and ileum show multifocal to diffuse, mild to severe villous atrophy, with necrosis and sloughing of degenerate enterocytes and a decreased villous-to-crypt ratio [72].

#### 3.6.3. Diagnostic Tools and Criteria

The main methods for laboratory diagnosis of PECs include polymerase chain reaction (PCR), immunofluorescence, immunohistochemistry, in situ hybridisation, electron microscopy and isolation in cell culture [62].

PEDV antigens can be detected in faeces using antigen capture ELISAs [58]. For the detection of PEDV RNA, the most widely used laboratory diagnostic methods are real-time RT-PCR assays and, given the higher diagnostic specificity of probe-based assays, are usually the best choice for molecular diagnosis even if various PCR-based design methods have also been widely used to diagnose co-infections of PECs [73]. For example, a TaqMan-probe-based multiplex real-time RT-qPCR assay has been designed to detect TGEV, PEDV, PDCoV and SADS-CoV in clinical samples [74].

Histopathology performed on fresh samples of small intestine (samples from acutely affected pigs euthanized and promptly necropsied to avoid the desquamation of enterocytes) should be considered among the first-choice diagnostic methods due to microscopic lesions highly compatible with enteric coronavirus infections. IF or IHC are useful diagnostic methods for the detection of PEDV antigens in paraffin-embedded sections of intestinal tissues [75].

Isolation of field strains of PEDV from intestinal contents/homogenates or faeces is conducted in Vero cells or in other cell types. Vero cells are the most widely used cell line for the isolation and propagation of PEDV [58].

Available serological methods for indirect diagnosis of PECs are not suitable for rapid and early diagnosis [76].

### 3.7. Enteric Salmonellosis

#### 3.7.1. Aetiology and Clinical Presentation

Salmonellosis in pigs occurs mainly in weaned pigs raised under intensive conditions. Suckling pigs can be infected, while the disease is infrequently observed, presumably due to protection given by lactogenic immunity. *Salmonella* Typhimurium and its monophasic variant *S*. 1,4,[5],12:i:- have worldwide distribution [77] and cause clinically indistinguishable diseases characterised by [12]:(1)fever;(2)inanition;(3)decreased feed intake;(4)yellow watery diarrhoea that may contain blood and mucous (especially in the later stages);(5)dehydration.

Dehydration and hypokalemia, which are established after several days of diarrhoea, are the cause of death. Disease most commonly develops in pigs with concurrent debilitating conditions, such as poor hygiene, that allow exposure to a high number of bacteria, or in immunologically naïve pigs.

Mortality is low and recovered pigs may remain as carriers, eliminating Salmonella intermittently for at least 5 months. The organism persists in [12]:(1)tonsils;(2)lower intestinal tract;(3)submandibular and ileocolic lymph nodes.

Some pigs may show wasting or may develop rectal strictures. In fact, *S*. Typhimurium causes ulcerative proctitis which may have a defective healing, resulting in obstipation, distention of the abdomen, anorexia, emaciation and soft faeces [12,77].

#### 3.7.2. Pathological Changes

*S.* Typhimurium causes necrotic enterotyphlocolitis generally involving the ileum, cecum and spiral colon and occasionally the descending colon and rectum, with formation of a diphtheritic membrane on the mucosal surface (Figure 18). Systemic dissemination and septicaemia are rare [12].

The affected intestinal segments involved typically have:(1)thickened oedematous wall;(2)red mucosa with a granular appearance;(3)multifocal or coalescing mucosal erosions;(4)presence of adherent fibrino-necrotic debris on the mucosa;(5)enlarged and congested mesenteric lymph nodes.

As reported above, the ulcerative proctitis of ischemic origin determine the rectal stricture described in cases of Salmonellosis.

Microscopic lesions, mainly observed in the cecum and spiral colon, are characterised by [12]:(1)focal to diffuse necrosis of crypt and enterocytes;(2)infiltration of neutrophils, macrophages and lymphocytes in the lamina propria and submucosa;(3)presence of fibrin thrombi in capillaries of the lamina propria;(4)less frequently, fibrin thrombi can be detected in larger vessels in the submucosa;(5)necrosis, with ulcerative lesions involving the mucosa, the submucosa and lymphoid patches.

Necrosis of lymphoid patches may be observed in the acute disease. Lymphoid hypertrophy or regenerative hyperplasia is instead more common in the subsequent phases of the disease. Regional lymph nodes can show focal necrosis, oedema and contain neutrophils and macrophages in the sinuses.

#### 3.7.3. Diagnostic Tools and Criteria

The compatible clinical signs and lesions must be confirmed by isolation of Salmonella and its identification for a definitive diagnosis of salmonellosis. Culture of enlarged mesenteric lymph nodes is of higher diagnostic value for enteric salmonellosis. Culture of faeces or intestinal mucosa using selective media, with or without enrichment, cannot allow for differentiation between infection (subclinical infection) and disease. This is further complicated by the possibility that pigs shed different Salmonella serotypes. It is important to highlight that the disease diagnosis cannot be based exclusively on the isolation of Salmonella by culture from the intestinal contents alone and must be always supported by the detection of appropriate lesions [77] (Figure 19).

Culture is typically followed by serotyping to confirm the serotype involved. PCR assays are increasingly available, often with serotype-level specificity that can reduce the time to the final aetiological diagnosis. Histologic examination of the affected intestine to differentiate intestinal salmonellosis from proliferative enteropathy and swine dysentery is of high diagnostic value.

PCR-based detection of salmonellae does not constitute diagnosis of salmonellosis since this assay can detect DNA from dead Salmonella and the organism may be present without causing clinical disease.

Serology is performed using ELISA tests based on surface antigens such as LPS or mixed antigens (LPS and antigens from several serotypes). These tests are useful for herd screening, while, due to a lack of specificity and sensitivity, they are not reliable for individual animal diagnosis. ELISA tests using meat juice sampled at slaughter have been used in Denmark and other European countries to categorise the level of Salmonella infection in pig herds [78].

### 3.8. Proliferative Enteropathy

#### 3.8.1. Aetiology and Clinical Presentation

Proliferative enteropathy (PE), caused by the obligate intracellular bacterium *Lawsonia intracellularis*, is an infectious disease also known as ileitis. The disease is characterised by the intestinal crypt epithelial cell proliferation which causes the thickening of the intestinal mucosa [79].

Three clinically different forms of PE are usually observed: (1) acute, (2) chronic and (3) subclinical (Figure 20):(1)The acute form, or proliferative haemorrhagic enteropathy (PHE), can be usually observed in young adult pigs 4–12 months of age. Clinical presentation is mainly characterised by sudden death associated with anaemia and haemorrhagic diarrhoea. In severe cases pigs may exsanguinate prior to the development of diarrhoea, and the only other clinical sign may be pallor [80]. However, in more prolonged cases, melena or haematochezia can be observed [79].(2)The chronic form, generally affecting 6–20 week old pigs, is the most common manifestation of PE, also known as porcine intestinal adenomatosis (PIA). Clinical signs, usually from mild to moderate, are characterised by non-haemorrhagic, grey or green, loose-to-watery diarrhoeic faeces associated with anorexia. Growth retardation may be observed despite normal feed intake. Infections by opportunistic bacteria can complicate chronic cases, resulting in necrotic enteritis or persistent diarrhoea, sometimes with liquid faeces containing fibrino-necrotic casts. In uncomplicated PE pigs usually recover about 5 weeks after the onset of clinical signs. In these cases, however, a reduction of pigs’ average daily weight gain and feed efficiency has been described [79].(3)The sub-clinical form is characterised by infected pigs with normal faeces, reduced weight gain, and less severe intestinal proliferative histological lesions, compared with the PIA.

**Figure 20 animals-13-00338-f020:**
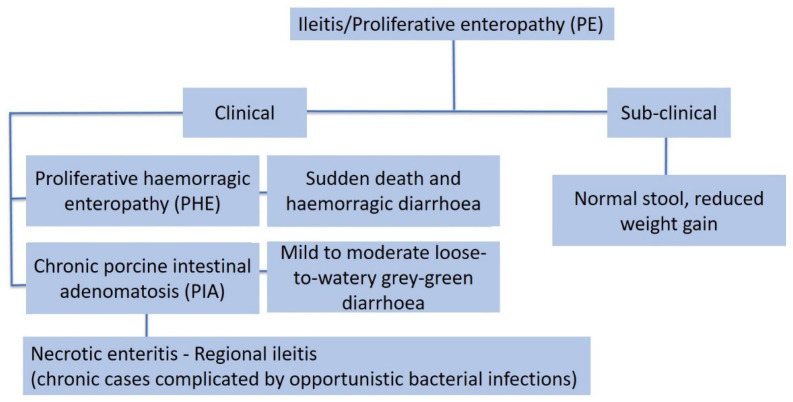
Clinical presentation of proliferative enteropathy (modified from D’Annunzio et al., 2021) [81].

#### 3.8.2. Pathological Changes

Macroscopic lesions associated with *L. intracellularis* infection are typically localised in the terminal ileum, from where they may extend distally to the colon and cecum, or proximally to the distal jejunum [82].

In the acute haemorrhagic form (PHE) (Figure 21), the wall of the ileum is thickened and it is common to find one or more formed blood clots and fibrino-necrotic debris in the intestinal lumen [80].

In mild cases of chronic PE (PIA), gross lesions are frequently observed in the terminal ileum (10 cm cranial to the ileocecal valve) [79]. The mucosa is thickened, corrugated or with a cerebriform appearance. In some cases, there is focal necrosis, with superimposed yellow or grey deposits of fibrin and cellular debris [83].

Secondary infection may lead to necrotic enteritis, characterised by coagulative necrosis and fibrin deposition. If the pig survives, the necrotic tissue is replaced by granulation tissue. This lesion is known as regional ileitis (Figure 22) [84].

The microscopic lesions of PE are very characteristic and generally pathognomonic (Figure 23) and include:(1)hyperplasia of crypt enterocytes with the formation of elongated, dilated and dysplastic crypts;(2)increased mitoses;(3)reduction in, or complete loss of, goblet cells.

**Figure 23 animals-13-00338-f023:**
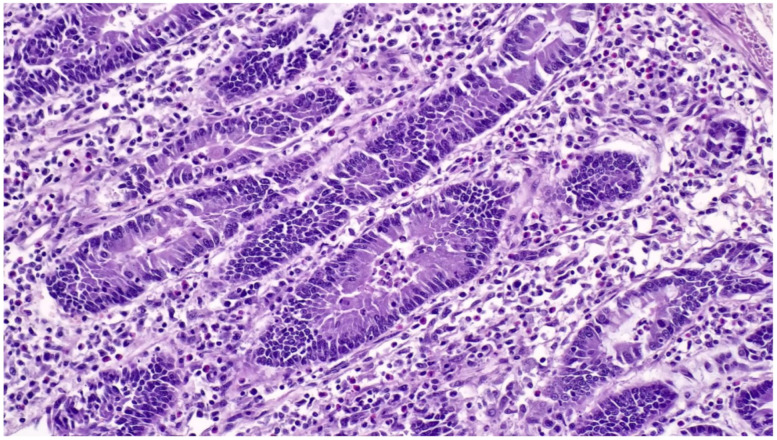
Ileum of pig affected by PE shows a hyperplastic epithelium of crypts (Haematoxylin and Eosin stain; 40×).

Upon recovery, the goblet cells reappear in the crypts. Inflammation is minimal or absent and can be observed only in old lesions or in cases complicated by opportunistic infections [85].

In cases of necrotic enteritis, the superficial mucosa shows coagulative necrosis with fibrinous exudate while infected crypts by *L. intracellularis* are located in the deepest and most intact parts of the intestinal mucosa.

Histological lesions in acute haemorrhagic cases (PHE) are characterised by mucosal lesions similar to those described for PE combined with fibrino-haemorrhagic clots in the lumen. *L. intracellularis* organisms mixed with necrotic-haemorrhagic debris can be observed on the mucosal surface and in the lumen of intestinal crypts [79].

#### 3.8.3. Diagnostic Tools and Criteria

The confirmatory diagnosis of PE requires the observation of characteristic macroscopic and microscopic lesions, combined with the detection of *L. intracellularis*. The gold standard is immunohistochemical staining using specific antibodies in order to demonstrate *L. intracellularis* in PE lesions (Figure 24) [79]. Negative IHC should be interpreted with caution due to the inexact segmental distribution of the PE lesions. For this reason, several intestinal segments should be evaluated in order to increase the probability of detecting *L. intracellularis* organisms within typical microscopic lesions [79].

Immunofluorescence-based tests can be applied for pathogen identification in tissues, but their routine use is limited by the need for fluorescence microscopy [80].

In situ hybridisation can be used to detect *L. intracellularis* DNA in typical lesions, but is uncommonly used for routine diagnostics due to the higher cost compared to IHC [80].

Another detection method for *L. intracellularis* in tissue sections is the Warthin–Starry silver stain technique. With this technique, *L. intracellularis* appears as a dark, curved organism in the apical cytoplasm of the enterocytes. This stain, however, does not confirm the identity of *L. intracellularis* and shows low specificity, reproducibility and repeatability [86].

The sensitivity of PCR techniques, such as standard PCR, real-time PCR, and loop-mediated isothermal amplification (LAMP), offer potentially rapid and sensitive ways to demonstrate the presence of *L. intracellularis* [80]. Many PCR assays can detect as few as 100 organisms per gram of faeces. The mere demonstration of *L. intracellularis* in faeces by PCR, however, cannot be used to confirm a diagnosis of PE [79].

The levels of *L. intracellularis* faecal shedding have been correlated with diarrhoeic disease and growth performance [79]. Ct values < 20, detected using a semi-quantitative real-time PCR, were predictive of finding lesions of PE and the detection of *L. intracellularis* by IHC, CT values > 30 were associated with a negative value for these variables, while CT values between 20 and 30 needed the interpretation within the clinical context, considering also other diseases affecting the pigs sampled [87]. The number of *L. intracellularis* excreted per gram of faeces, determined by qPCR, is usually between 10^4^ and 10^8^. This number varies depending on whether pigs are clinically or sub-clinically affected with PE, with clinically affected pigs excreting more than 10^7^ *L. intracellularis* per gram of faeces [88]. Antibodies against *L. intracellularis* can be detected in serum 2 weeks after infection, they peak at 3–4 weeks and remain detectable for up to 13 weeks post-infection [80]. Several ELISAs have been developed and are commercially available for the detection of *L. intracellularis* antibodies in serum. Serology is performed mainly for monitoring purposes rather than for diagnostic investigation. Diagnostic tools and criteria for the diagnosis of PE in pigs are reported in Figure 25.

### 3.9. Swine Dysentery and Porcine Colonic Spirochetosis

#### 3.9.1. Aetiology and Clinical Presentation

Swine dysentery (SD) is a severe muco-haemorrhagic colitis caused by infection with strongly haemolytic *Brachyspira* species, especially *B. hyodysenteriae,* but also *B. hampsonii* and *B. suanatina* [89]. *B. pilosicoli* is instead the aetiological agent of porcine intestinal spirochetosis (PIS) or porcine colonic spirochetosis (PCS).

The clinical presentation of SD is observed mainly in grower and finisher pigs and less frequently in weaner pigs, and is characterised by [89,90]:(1)at the onset: soft, yellow to grey diarrhoeic faeces, with large amounts of mucous and often flecks of blood and subsequently watery faeces containing blood, mucous and muco-fibrinous exudate (Figure 26A,B);(2)partial anorexia;(3)retarded growth rate;(4)fever (40–40.5 °C);(5)mortality rate of 50–90%.

**Figure 26 animals-13-00338-f026:**
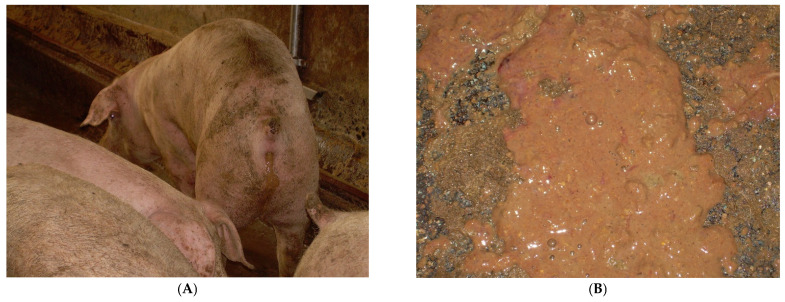
Pig with typical diarrhoea due to *B. hyodysenteriae* (**A**). Faeces are characterised by mucous, blood and muco-fibrinous exudate (**B**).

Severely affected pigs die due to dehydration, metabolic acidosis and hyperkalemia [91].

In farms where the disease is endemic, the clinical signs can reappear cyclically at intervals of 3–4 weeks, involving few pigs or large groups [89]. Possible relapses of the disease can be observed after the suspension of antimicrobial treatment or following the effect of various stressors, including moving to new pens, mixing with different animals, weighing, change in feed, overcrowding and/or extreme changes in environmental temperature [89].

PIS/PCS is characterised by green or brown watery to mucoid diarrhoea, lasting from 2 to 14 days, which then may recur. The consistency of faeces can change, resembling wet cement or porridge [92]. Pigs with recurrent diarrhoea may show significant loss of body condition, poor feed conversion rates, and delays in reaching market weight [93]. Several enteric diseases may be confused with SD, and should be included in the differential diagnosis (Table 2) [89].

#### 3.9.2. Pathological Changes

In acute SD lesions are limited to the cecum and descending colon. The spiral colon usually shows the most severe lesions [94]. Typical gross lesions in the first stage of disease include [89]:(1)hyperemia;(2)oedema of the large intestinal wall, mesentery and mesenteric lymph nodes;(3)oedematous intestinal mucosa (Figure 27A);(4)mucous and fibrin, mixed with blood cover the intestinal mucosa (Figure 27B);(5)loose to watery stools.

**Figure 27 animals-13-00338-f027:**
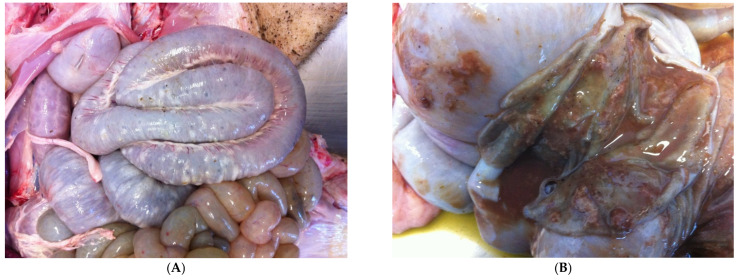
Colon of pig affected by SD shows oedema of the intestinal wall (**A**) and intestinal mucosa showing mucous, fibrin and blood (**B**).

Thick pseudomembranes consisting of fibrin, mucous and blood are observed as the disease progresses, while a thin layer of fibrino-necrotic exudate characterises lesions described in chronic cases [91].

As mentioned above, porcine intestinal spirochetosis is characterised by gross lesions limited to the cecum and colon. The mucosa is generally congested, hyperemic, showing catarrhal exudation and/or multifocal erosive or necrotic lesions [92].

Microscopic lesions of SD are limited to the large intestine. In the acute phase of disease, spirochetes are most numerous in the lumen and within superficial crypts [89]. Early microscopic lesions of SD include [89]:(1)superficial mucosal necrosis;(2)neutrophilic infiltration of the lamina propria;(3)crypt elongation;(4)haemorrhage;(5)goblet cells hyperplasia.

As the disease progresses, fibrin, mucous and cellular debris accumulate in mucosal crypts and on the surface of the intestinal mucosa [89].

In cases of PIS/PCS the mucosa is thickened, oedematous and occasionally hyperemic. Inside the crypts *B. pilosicoli* organisms can be observed as well as the accumulation of neutrophils (crypt abscesses), mucous and cellular debris. The lamina propria is usually infiltrated by neutrophils and lymphocytes [92].

#### 3.9.3. Diagnostic Tools and Criteria

Selective culture for *Brachyspira* spp. from clinical samples (colon and faeces) provides a high degree of diagnostic sensitivity for SD [88] and allows the demonstration of a strongly beta-haemolytic *Brachyspira* spp. [89] (Figure 28).

Faeces and colonic mucosa of clinically affected pigs, untreated with antimicrobials and in the acute phase of the disease, contain large numbers (10^8^–10^9^/g) of *Brachyspira* spp. Subclinically infected pigs intermittently shed organisms at detectable levels (>10^3^ cells/mL contents) in their faeces and for these reason are not reliable for diagnostic investigation [89,90].

PCR assays or, as an alternative, MALDI-TOF MS, are generally used to identify the strains obtained from primary cultures [91,95]. Direct PCR from clinical samples may lack sensitivity relative to microbial culture [96].

In situ hybridisation assays targeting the rRNA of *B. hyodysenteriae*, *B. pilosicoli* and *B. hampsonii* have been reported [97,98].

Serologic assays have been developed over time with improved specificity to detect circulating antibodies against *B. hyodysenteriae*-specific outer membrane lipoproteins; however, these assays have not yet found broad application [91].

## 4. Conclusions

A correct diagnostic approach is fundamental for disease’s control. Diagnostic pathways require appropriate sampling for isolation or the demonstration of pathogens, for the evaluation of characteristic microscopic lesions and for the co-localisation of specific antigens within observed microscopic lesions. Enteric diseases are among the main health and economic problems of pig production. At the same time, it is very important to remember that most agents affecting the intestinal tract in swine are ubiquitous and considered part of the normal microbiota. In many cases, the isolation of the pathogen alone does not imply a definitive diagnosis, which is only possible strictly respecting the diagnostic criteria defined for every single disease. Table 3 summarises the most important clinical and pathological findings and the diagnostic tools of the main enteric diseases of swine.

## Figures and Tables

**Figure 1 animals-13-00338-f001:**
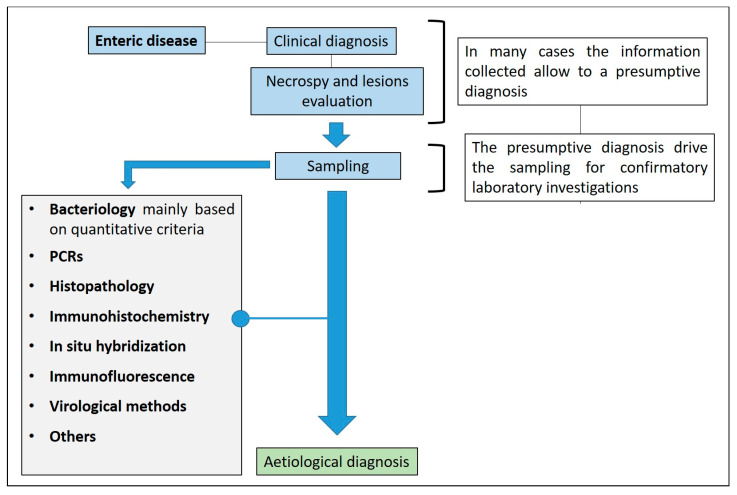
Enteric disease and diagnostic algorithm.

**Figure 2 animals-13-00338-f002:**
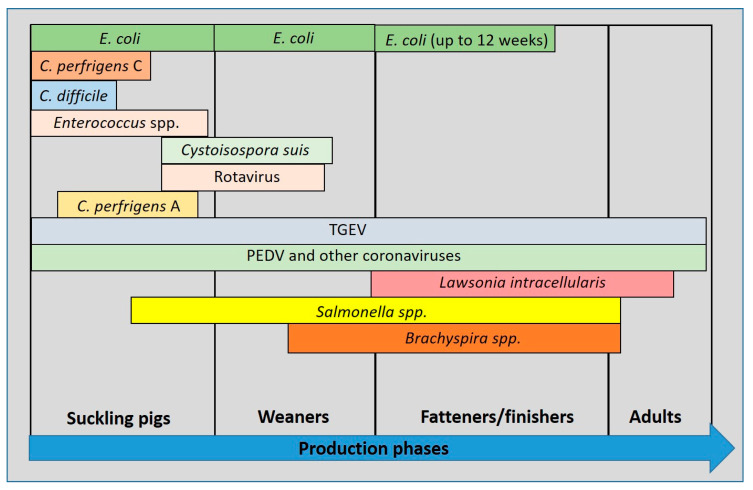
Incidence of pathogens in enteric disease in pigs related to age (modified from Ségales et al., 2013) [6].

**Figure 3 animals-13-00338-f003:**
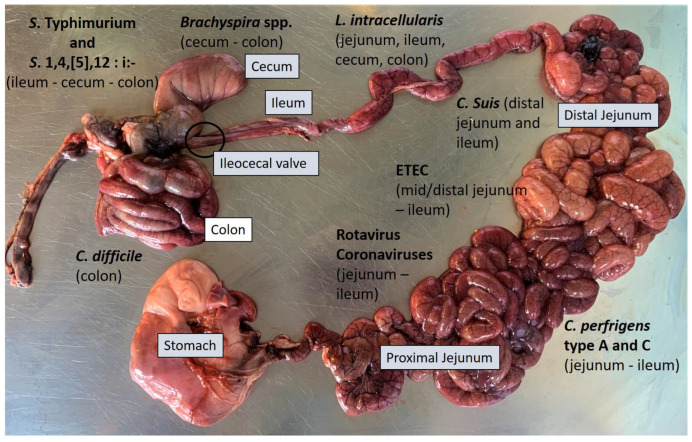
Intestine of pig. Main distribution of the aetiological agents and related lesions.

**Figure 4 animals-13-00338-f004:**
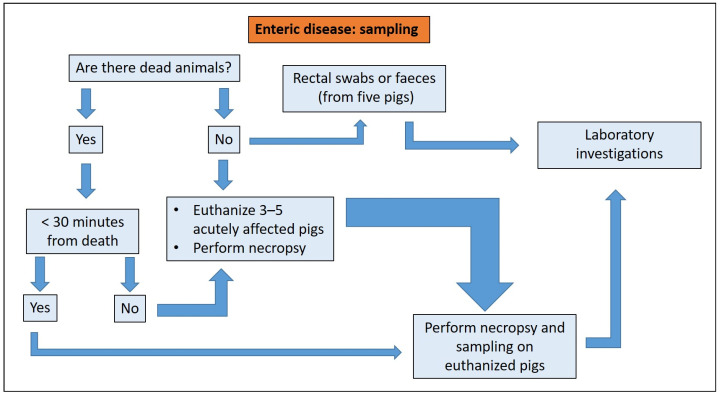
Enteric disease: sampling criteria in the diagnostic pathway.

**Figure 5 animals-13-00338-f005:**
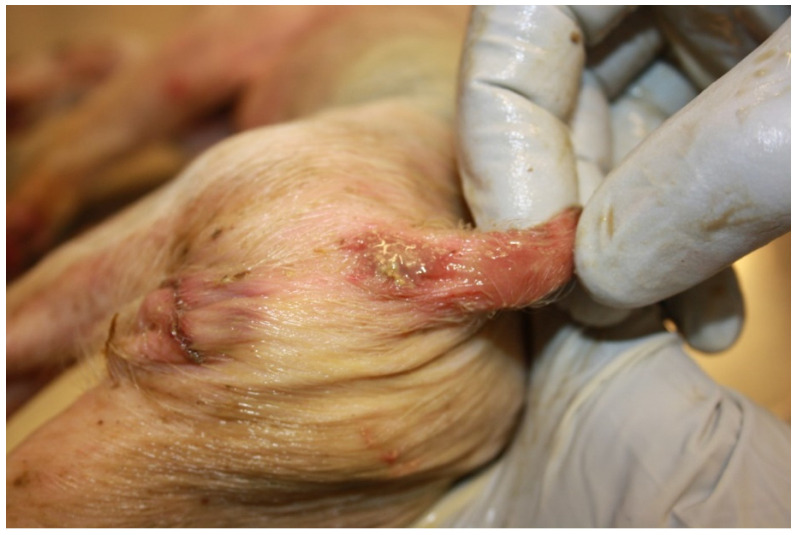
Suckling piglet suffering from colibacillosis. Characteristic wet hair coat and inflammation of anus.

**Figure 6 animals-13-00338-f006:**
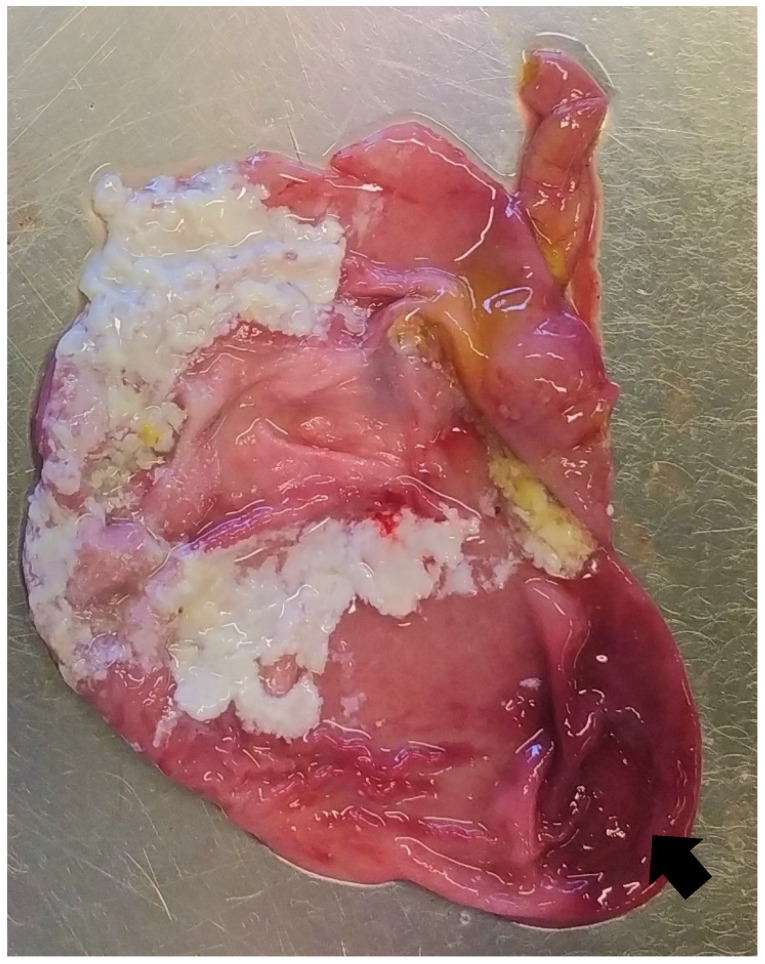
Suckling piglet affected by colibacillosis. The stomach shows hyperaemia of the fundus (arrow).

**Figure 7 animals-13-00338-f007:**
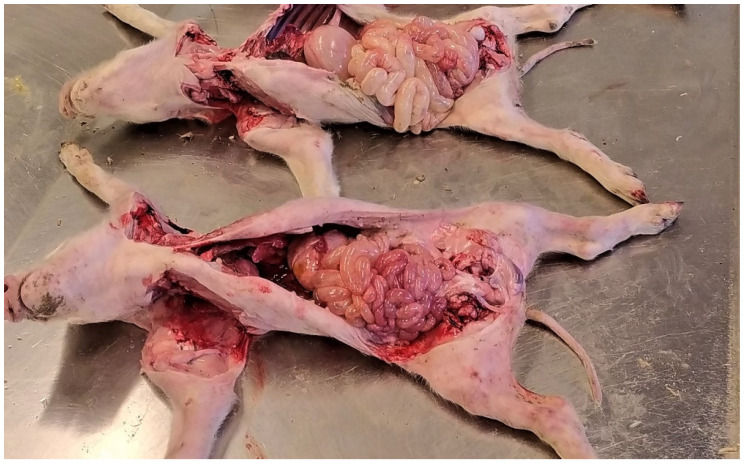
Suckling piglets suffering from colibacillosis. The small intestine is dilated, meteoric and hyperaemic.

**Figure 8 animals-13-00338-f008:**
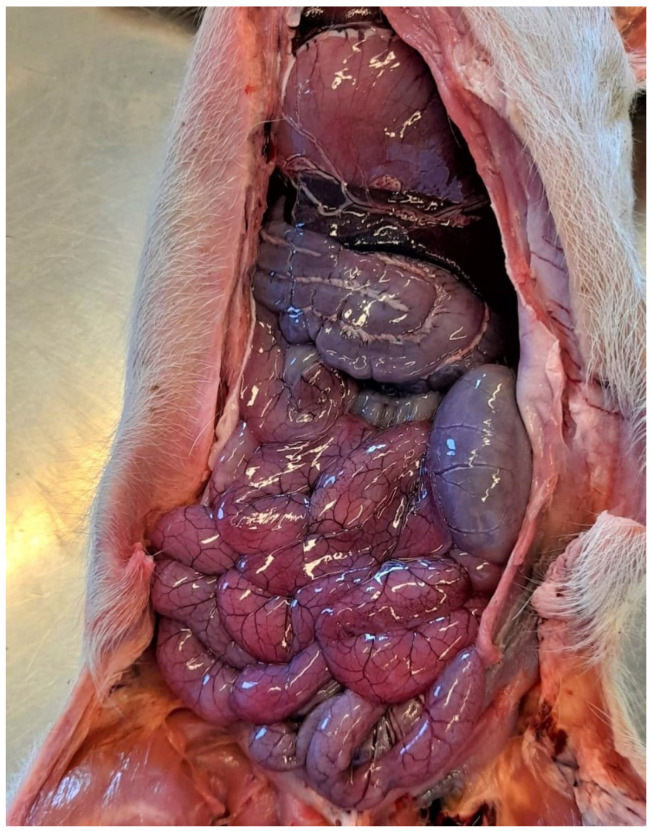
Pig 45 days old with colibacillosis due to F4, STa, STb. The intestine is dilated and hyperaemic.

**Figure 9 animals-13-00338-f009:**
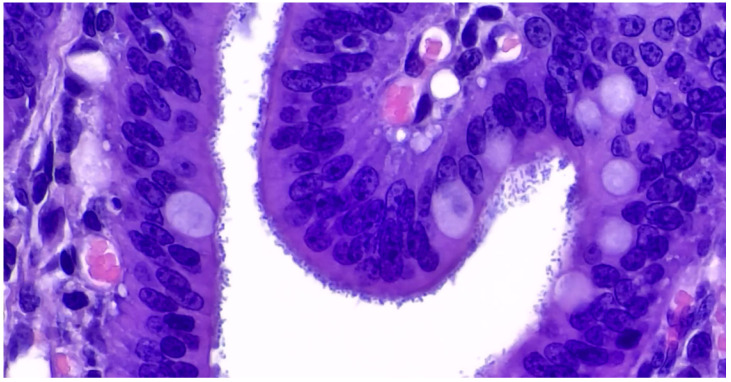
Pig 45 days old. Colibacillosis due to F4, STa, STb ETEC strain. Histological evaluation of jejunum shows bacterial layers attached to the brush border enterocytes (Haematoxylin and Eosin stain, 60×).

**Figure 10 animals-13-00338-f010:**
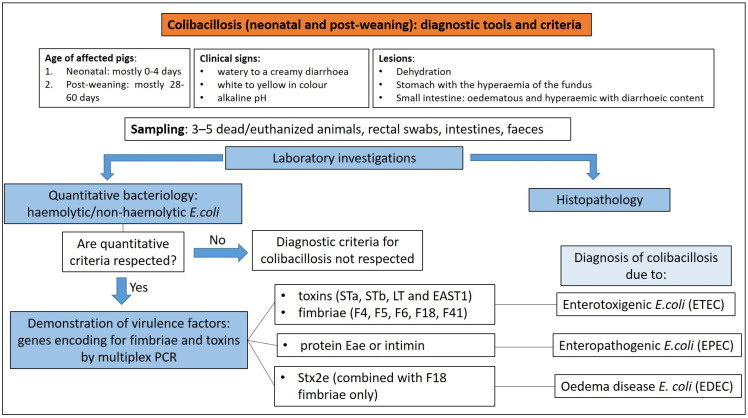
Diagnostic algorithm for the diagnosis of colibacillosis.

**Figure 11 animals-13-00338-f011:**
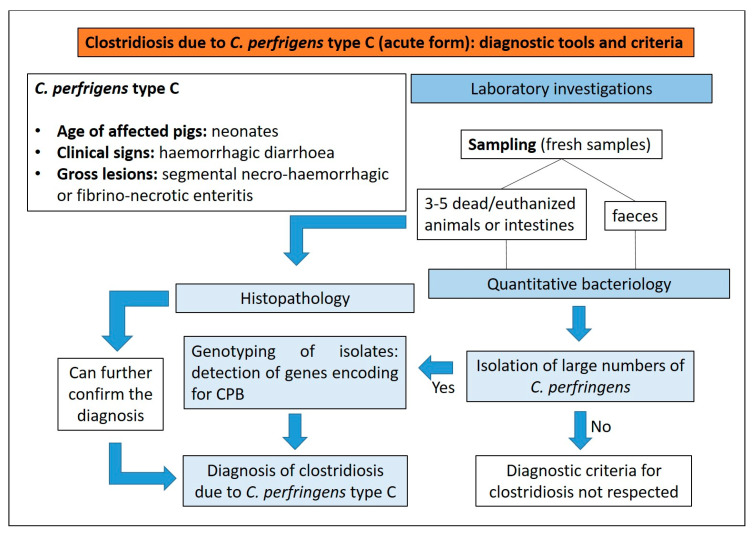
Diagnostic algorithm for the diagnosis of clostridiosis due to *C. perfrigens* type C in pigs.

**Figure 13 animals-13-00338-f013:**
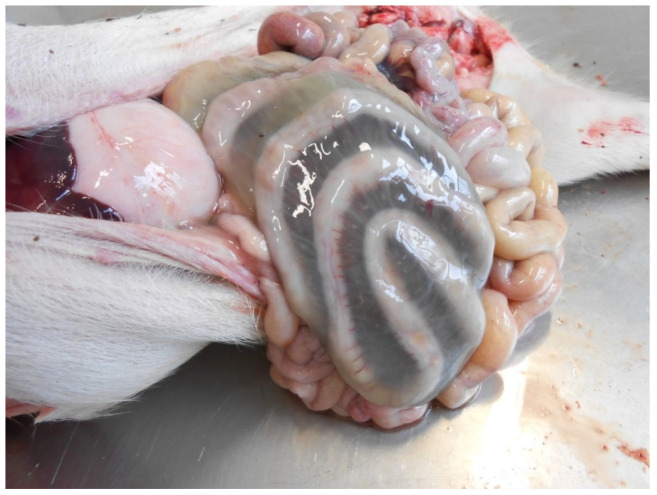
Piglet 5 days of age. Marked oedema of the mesocolon in a case of clostridiosis due to *C. difficile*.

**Figure 15 animals-13-00338-f015:**
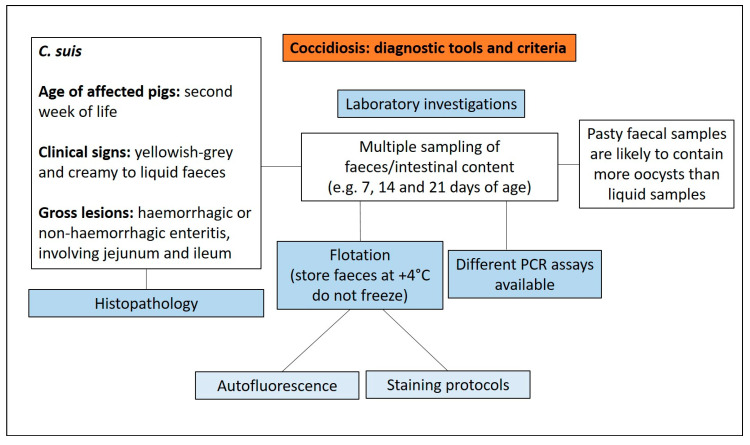
Diagnostic algorithm for the diagnosis of coccidiosis.

**Figure 16 animals-13-00338-f016:**
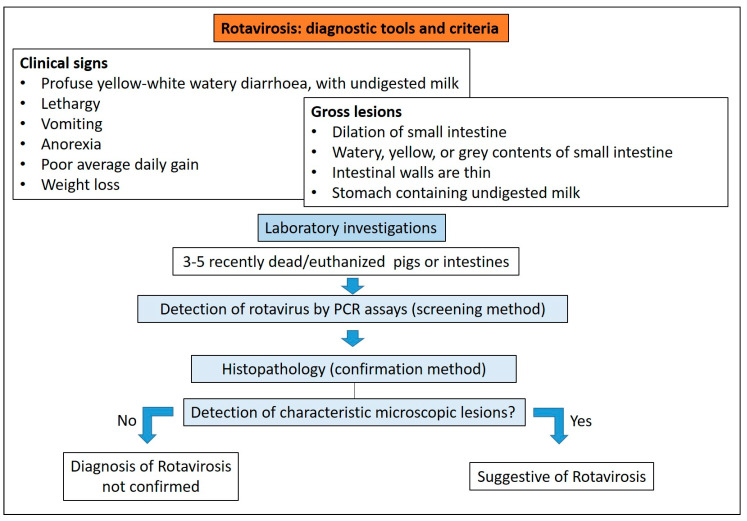
Diagnostic algorithm for the diagnosis of rotavirosis in pigs.

**Figure 17 animals-13-00338-f017:**
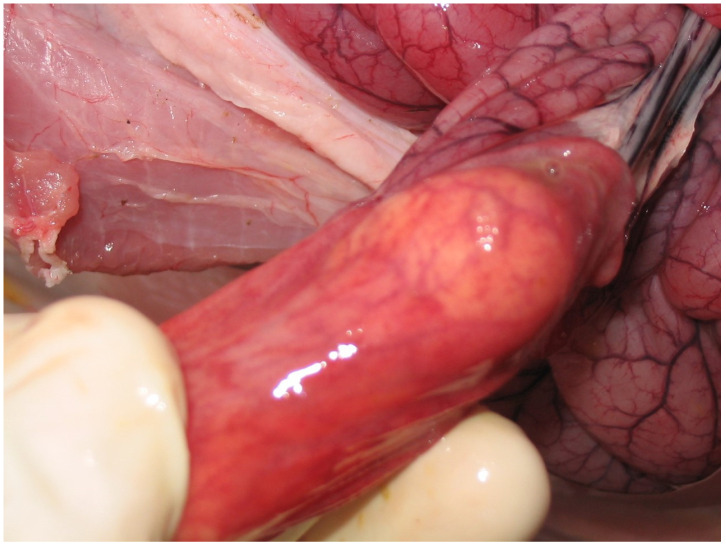
Extreme thinning of the small intestine of a 45-day-old pig affected by PED.

**Figure 18 animals-13-00338-f018:**
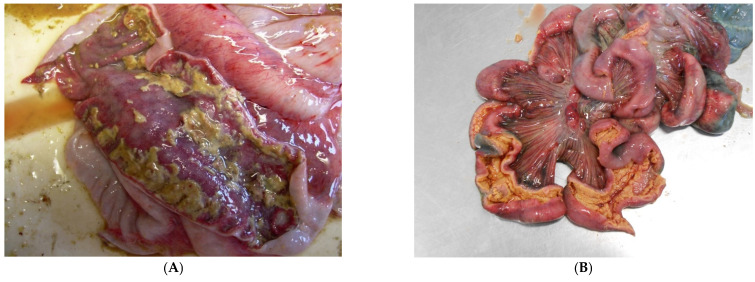
Fibrinous enteritis with involvement of colon (**A**) and small intestine (**B**) in pigs infected with *S.* Typhimurium.

**Figure 19 animals-13-00338-f019:**
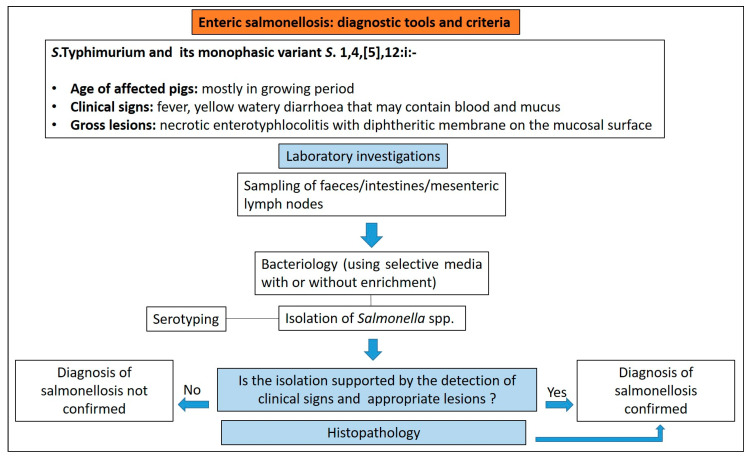
Diagnostic algorithm for the diagnosis of salmonellosis in pigs.

**Figure 21 animals-13-00338-f021:**
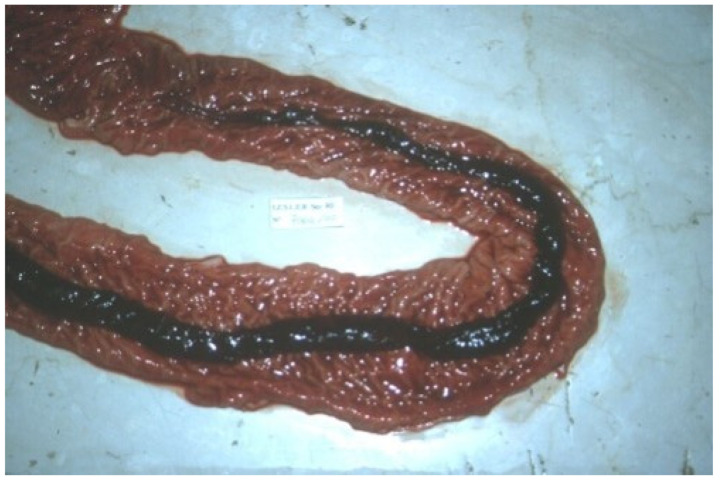
Ileum of pig affected by PHE with a thickened wall and a blood clot in the lumen.

**Figure 22 animals-13-00338-f022:**
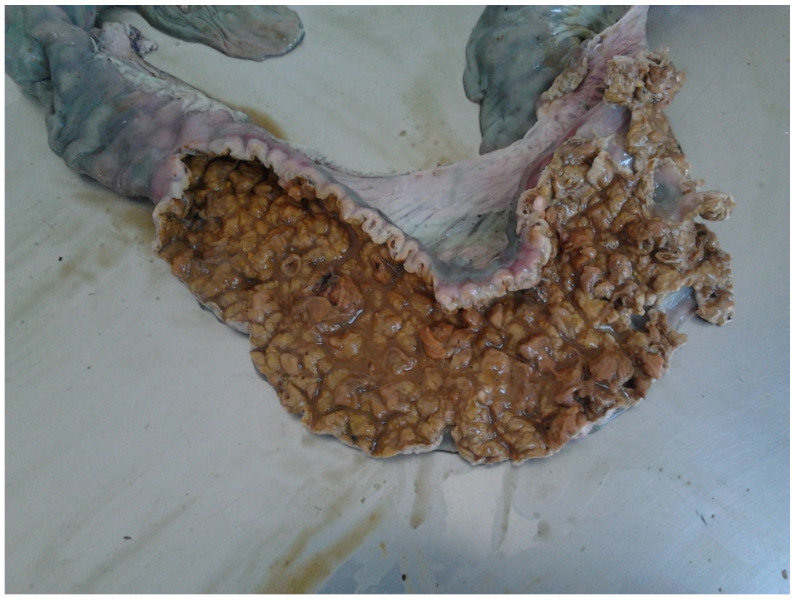
Ileum of pig. The picture shows a PIA complicated by secondary bacterial infections leading to necrotic enteritis, also known as regional ileitis.

**Figure 24 animals-13-00338-f024:**
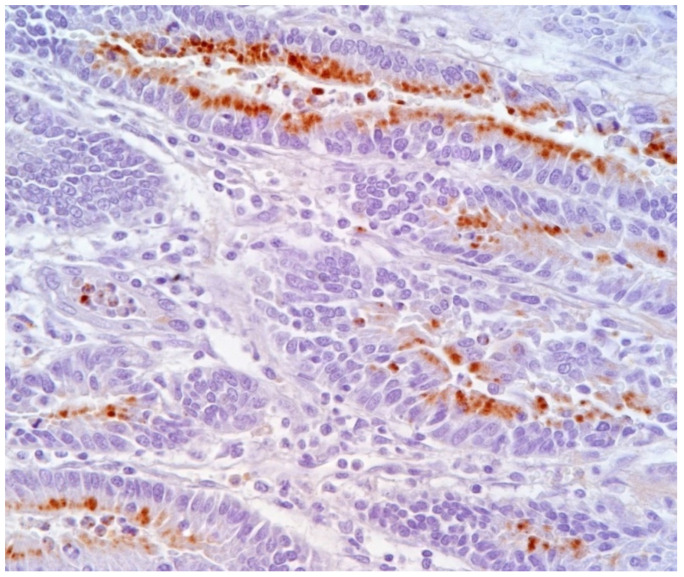
Ileum of pig affected by PE. *L. intracellularis* organisms in the apical cytoplasm of hyperplastic epithelium lining intestinal glands (IHC, 40×) (Courtesy of Prof. G. Sarli).

**Figure 25 animals-13-00338-f025:**
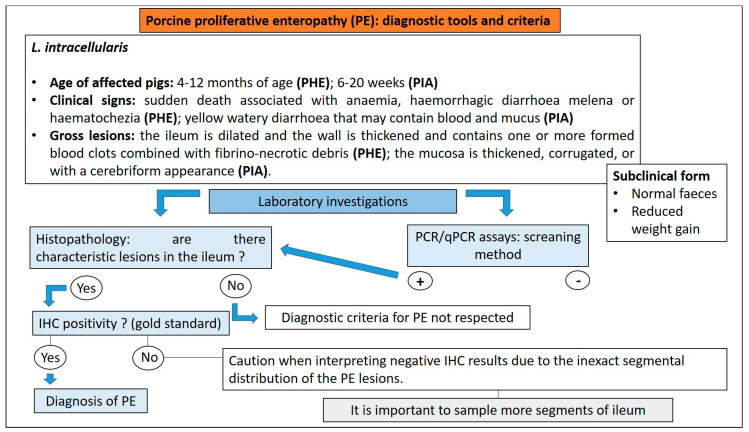
Diagnostic algorithm for the diagnosis of PE in pigs.

**Figure 28 animals-13-00338-f028:**
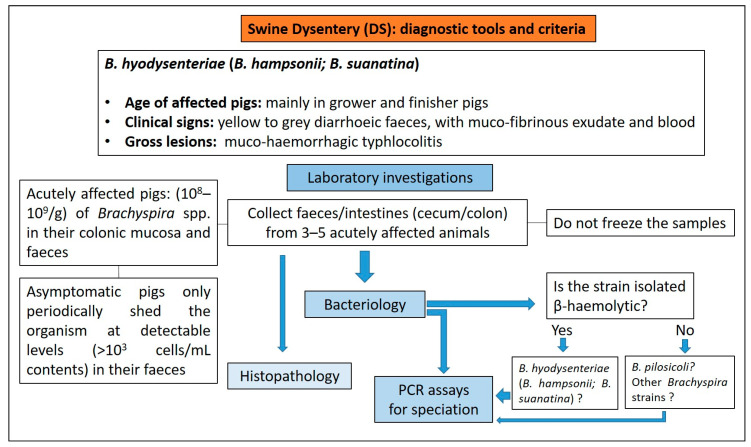
Diagnostic algorithm for the diagnosis of SD in pigs.

**Table 1 animals-13-00338-t001:** Enteric disease: specimen collection for histopathology (Modified from Arruda and Gauger, 2019) [4].

Tissue/Sample	Specimen Collection
Lymph node	Mesenteric—1 cm thickness
Tonsils	Half of a tonsil
Spleen	1 cm thickness
Liver	1 sample 2 × 2 × 0.5 cm
Kidney	Half of a kidney, 0.5 cm slice through the centre
Stomach	3 × 3 × 3 cm piece 1 cm thickness
Jejunum	Three sections, 2 cm long
Ileum	Three sections, 2 cm long
Spiral colon	Three sections, 2 cm long

**Table 2 animals-13-00338-t002:** SD and differential diagnosis [89].

Disease/Aetiological Agent	Main Clinical and Anatomopathological Characters
Proliferative enteropathy (PE) *L. Intracellularis*	Clinically resembles SD, but SD does not affect the small intestine
Enteric salmonellosis *S*. Typhimurium and its monophasic variant	Clinical signs and gross lesions can be similar. Parenchymatous organs and lymph nodes necrosis, fibrinous and ulcerative lesions in the small intestine not observed in SD
Trichuriasis *Trichuris suis*	Large numbers of *Trichuris suis* in the cecum
Gastric ulcers and other haemorrhagic conditions	Digested blood in the faeces, “tarry” appearance; the large intestine has no lesions.
Porcine intestinal spirochetosis *B. Pilosicoli*	The differential diagnosis is difficult, being similar to mild cases of SD

**Table 3 animals-13-00338-t003:** Clinical and pathological findings and diagnostic tools of the main pig enteric diseases.

Disease/Aetiological Agent	Age	Clinical and Pathological Findings	Diagnostic Tools
Neonatal and Post-weaning Colibacillosis *E.coli* (ETEC)	Neonatal: mostly 0–4 days Post-weaning: mostly 28–60 days	Watery/creamy diarrhoea, white to yellow in colour Alkaline pH Small intestine: oedematous and hyperaemic with diarrhoeic content (characteristic smell) Stomach with the hyperaemia of the fundus	Quantitative bacteriology Typing of isolates by PCR Histopathology
Clostridiosis *C. perfrigens* type C	Neonates (until 3 weeks of age)	Haemorrhagic diarrhoea Segmental necro-haemorrhagic or fibrino-necrotic enteritis	Quantitative bacteriology Typing/toxin identification Histopathology
Clostridiosis *C. perfrigens* type A	Neonates/suckling piglets	Lack of clear criteria for definitive diagnosis Absence of characteristic clinical and pathological findings	Quantitative bacteriology Typing/toxin identification Histopathology
Clostridiosis *Clostridioides difficile*	1–7 days of life	Pasty-to-watery yellowish faeces Mesocolonic oedema Necro-suppurative or erosive/ulcerative colitis and typhlocolitis	Bacteriology Toxin identification Enzyme immunoassays (for toxins detection) Histopathology
Coccidiosis *Cystoisospora suis*	Commonly in the second week of life	Yellowish-grey and creamy to liquid faeces Haemorrhagic or non-haemorrhagic enteritis, involving the jejunum and ileum	Microscopic evaluation after flotation Autofluorescence/staining protocols PCR Histopathology
Rotavirosis Rotavirus	Commonly in 2 to 6 weeks old pigs	Profuse yellow-white watery diarrhoea (acid pH), with undigested milk Lethargy, vomiting, anorexia Poor average daily gain, weight loss Small intestine dilatation and intestinal thinning	PCR Histopathology
Coronaviruses (PECs)	All	Empty stomach Small intestine was thinned and congested	PCR Histopathology
Salmonellosis (*Salmonella* Typhimurium and its monophasic variant S. 1,4,[5],12:i:-)	Mostly in growing period	Fever, yellow watery diarrhoea with blood and mucous Necrotic enterotyphlocolitis with diphtheritic membrane on the mucosal surface	Bacteriology Serotyping Histopathology
Proliferative enteropathy *Lawsonia intracellularis*	4–12 months of age (PHE); 6–20 weeks (PIA)	Sudden death, anaemia, haemorrhagic diarrhoea, melena or haematochezia (PHE) Yellow watery diarrhoea (blood, mucous) (PIA) Ileum dilatation, the wall is thickened with one or more formed blood clots (PHE) The mucosa is thickened, corrugated, cerebriform (PIA)	PCR/qPCR assays Histopathology Immunohistochemistry (gold standard)
Swine dysentery (SD) *Brachyspira hyodysenteriae* (*B. hampsonii; B. suanatina*) Intestinal spirochetosis (PIS) *B. pilosicoli*	Mainly in grower and finisher pigs	Yellow to grey diarrhoeic faeces, with muco-fibrinous exudate and blood (SD) Muco-haemorrhagic typhlocolitis (SD) Green or brown watery to mucoid diarrhoea (PIS)	Bacteriology Typing by PCR Histopathology

## Data Availability

Data are not available.

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
