# Peer review of "Diagnostic Approach to Enteric Disorders in Pigs"

_animals, 2023, doi:10.3390/ani13030338_

Round 1

Reviewer 1 Report

General comments

This is a comprehensive review regarding the diagnosis of enteric diseases in pigs. It is very well written and it will bring substantial updated information for the readers. A few main point to be modified before final acceptation would be the following:

1.       I believe the images placed as schemes are in fact “Figures”, so pictures or schemes should all be considered figures and numbered accordantly from 1 to 28.

2.       In page 2, line 9, the name of the genus “Clostridium” is incorrect. The correct name is “Clostridioides difficile”.

3.       In Figure 2 (name in the manuscript as Scheme 2), Escherichia coli in Fatteners/finishers should state “Escherichia coli, up to 12 weeks” or reduce the length of the bar to indicate that E. coli would cause problems only until around 12 weeks. In the same sense, reduce the bars of C perfringens type C and Clostridioides difficile as they will cause diarrhea in piglets up to 72 hours and 5 days, respectively. L. intracellularis may be involved with clinical cases of diarrhea in late nursery in Europe, as pigs leave to this site usually with 30 kg, but in the rest of the world the keep pigs in nursery up to 60-65 days (23-25 kg) it is VERY RARE to see any diarrhea due to L. intracellularis in that phase. Finally, the images of lesions in figure 2 are too small and it is impossible to see any detail. I recommend deleting these images.

4.       In figure 3 (former figure 1) L. intracellularis infection and lesions can be found in the jejunum, ileum, cecum and colon, so add these segments to the image.

5.       In figure 4, and throughout the manuscript is stated that segments of intestine from animals could be necropsied and samples collected for bacteriology and histology up to 4 hours after death. I know that you are referring to Segales et al 2013, but after 30 minutes the intestine start to had significant post mortem lesions such as detachment of enterocytes that will make histopathological interpretation very difficult. So, I would recommend to clarify in the text that tissue from euthanized animals are preferred to animals found dead for laboratory submissions. Modify Figure 4 accordantly, as you should perform the post mortem examination in animals found dead, but sample collection should come preferable from euthanized animals.

6.       At the beginning of page 5, it is stated that the ends of the fragments of intestine should be tied off. There is no specific justification for that, mainly for the small intestine that has less content and because bacteriological samples will be taken from the middle of the segment. In addition, all intestinal fragments should be kept in a different bag away from other tissues to avoid contamination. In fact, this sample collection clarification, despite of being obvious, should be stated in the text to inform about adequate tissue submission to the laboratory.

7.       In figure 7 (former figure 4) it is impossible to see edema in the image of intestines, so delete this word from the legend. The same applies to figure 8 (former figure 5). Edema is only verified when you open a tubular organ and see thickening of the intestinal wall with accumulation of material in the area supposed to be the submucosa.

8.       Delete the expression “… mild enteritis with…” in the first line of the last paragraph of page 8, as usually no typical inflammation of neutrophils infiltration is seemed in ETEC intestinal infection.

9.       In figure 10 (former scheme 4) and in many other figures the indication of “Histopathology” is followed by “(complementary investigation)”. My understanding is that all samples submitted to the laboratory will be tested by complementary exams. In addition, all submitted samples have to be screened by a pathologist before application of any complementary test or exam, and, at the end, all results from all exams come to the pathologist for the conclusion of the case based on the clinical history, gross and microscopic lesions. As a result, histology has a pivotal role for the diagnosis process, and not only a “complementary investigation”. Please, delete this expression (complementary investigation) all figures.

1 In page 11, second paragraph of section 3.2.1.3. Diagnostic tools and criteria, all Clostridium perfringens types produces a double zone of hemolysis, not only C. perfringens type C. Modify the sentence to make it clear.

1 In page 12, in the first line of the topic 3.2.1.1. Aetiology and clinical presentation, please, substitute the word “flora” by “microbiota”.

1In figure 12 (former scheme 6), make clear that the detection of CPB2 does not contribute at all for the diagnosis of C. perfringens type A, as demonstrated in many published studies. The exclusion of all other possible causes and the visualization of a myriad of bacilli in close contact to the enterocytes in the small intestine are the only indication of “POSSIBLE” indication of the C. perfringens type A involvement in the diarrhea.

1 In page 15, in the first line of the topic 3.3.1. Aetiology and clinical presentation, please, substitute the word “flora” by “microbiota”.

1In page 16, in the first line of section 3.4.1. Aetiology and clinical presentation, delete the word “Neonatal” as coccidiosis can only manifest after 5 days of age. It might give the false impression that it would cause diarrhea in very young piglets. Start the sentence with “Coccidiosis caused….”

1In figure 16 (former scheme 9), if you have histologic lesions indicative of viral infection and PCR positive for Rotavirus, this is confirmatory diagnostic for Rota and does not require any other direct detection of the virus by IHC or ISH. Sometimes these two last tests can be negative and the animal still Rotavirus positive due to exfoliation of enterocytes and reduction of infected cells. Modify the figure accordantly.

1Figure 25 (former scheme 12):  PHE is not associated with “fever”, please, delete this word from the figure.

1In figure 30, the item (5) of the list of Early microscopic lesions of SD include [89]: substitute “mucus secretion” by “goblet cells hyperplasia”.

1In page 31, in the second paragraph, I believe MALDI-TOF MS does not allow adequate  discriminatory differences among different Brachyspira spp. Please, confirm this information.

Author Response

Parma, 6st of January 2022

To the Editor of Animals,

Subject: Response to Reviewer’s comments on the manuscript – Diagnostic approach to enteric disorders in pigs - Manuscript ID: Animals-2117665. Type: Review

Authors: Andrea Luppi, Giulia D’Annunzio, Camilla Torreggiani, Paolo Martelli

Section: Pigs - Special Issue: Gastrointestinal Tract Health in Pigs

Dear Editor,

Thank you for assessing the manuscript “Diagnostic approach to enteric disorders in pigs" and for your positive response on its potential publication in Animals following the proposed revisions suggested by the reviewers. Thank you for the comments and review proposals that led to an improvement of the manuscript.

You will find below the responses to the Reviewer’s comments and specific changes implemented in the manuscript as a result of these comments. All changes made have been highlighted with 'tracked changes'.

I trust that the reviewed version of the manuscript, attached to this letter, will satisfy these comments and is now acceptable for publication in Animals.

Looking forward to hearing from you.

Yours faithfully,

Andrea Luppi

Corresponding author for manuscript submission (Animals-2117665).

Response to Reviewer 1 Comments

This is a comprehensive review regarding the diagnosis of enteric diseases in pigs. It is very well written and it will bring substantial updated information for the readers. A few main point to be modified before final acceptation would be the following:

Point 1:       I believe the images placed as schemes are in fact “Figures”, so pictures or schemes should all be considered figures and numbered accordantly from 1 to 28.

Response 1: Done

Point 2:       In page 2, line 9, the name of the genus “Clostridium” is incorrect. The correct name is “Clostridioides difficile”.

Response 2: Changed Clostridium in Clostridioides

Point 3:       In Figure 2 (name in the manuscript as Scheme 2), Escherichia coli in Fatteners/finishers should state “Escherichia coli, up to 12 weeks” or reduce the length of the bar to indicate that E. coli would cause problems only until around 12 weeks. In the same sense, reduce the bars of C perfringens type C and Clostridioides difficile as they will cause diarrhea in piglets up to 72 hours and 5 days, respectively. L. intracellularis may be involved with clinical cases of diarrhea in late nursery in Europe, as pigs leave to this site usually with 30 kg, but in the rest of the world the keep pigs in nursery up to 60-65 days (23-25 kg) it is VERY RARE to see any diarrhea due to L. intracellularis in that phase. Finally, the images of lesions in figure 2 are too small and it is impossible to see any detail. I recommend deleting these images.

Response 3: The changes in figure 2 have been done as addressed by the reviewer.

Point 4:       In figure 3 (former figure 1) L. intracellularis infection and lesions can be found in the jejunum, ileum, cecum and colon, so add these segments to the image.

Response 4: Done

Point 5:       In figure 4, and throughout the manuscript is stated that segments of intestine from animals could be necropsied and samples collected for bacteriology and histology up to 4 hours after death. I know that you are referring to Segales et al 2013, but after 30 minutes the intestine start to had significant post mortem lesions such as detachment of enterocytes that will make histopathological interpretation very difficult. So, I would recommend to clarify in the text that tissue from euthanized animals are preferred to animals found dead for laboratory submissions. Modify Figure 4 accordantly, as you should perform the post mortem examination in animals found dead, but sample collection should come preferable from euthanized animals.

Response 5: Figure 4 was modified according to the suggestion of the Reviewer. The concept about the sampling of death or euthanized animals was reported in the text.

Point 6:       At the beginning of page 5, it is stated that the ends of the fragments of intestine should be tied off. There is no specific justification for that, mainly for the small intestine that has less content and because bacteriological samples will be taken from the middle of the segment. In addition, all intestinal fragments should be kept in a different bag away from other tissues to avoid contamination. In fact, this sample collection clarification, despite of being obvious, should be stated in the text to inform about adequate tissue submission to the laboratory.

Response 6: The text was modified according to the advice of the Reviewer.

Point 7:      In figure 7 (former figure 4) it is impossible to see edema in the image of intestines, so delete this word from the legend. The same applies to figure 8 (former figure 5). Edema is only verified when you open a tubular organ and see thickening of the intestinal wall with accumulation of material in the area supposed to be the submucosa.

Response 7: Modified the legends as addressed.

Point 8:       Delete the expression “… mild enteritis with…” in the first line of the last paragraph of page 8, as usually no typical inflammation of neutrophils infiltration is seemed in ETEC intestinal infection.

Response 8: Done

Point 9:       In figure 10 (former scheme 4) and in many other figures the indication of “Histopathology” is followed by “(complementary investigation)”. My understanding is that all samples submitted to the laboratory will be tested by complementary exams. In addition, all submitted samples have to be screened by a pathologist before application of any complementary test or exam, and, at the end, all results from all exams come to the pathologist for the conclusion of the case based on the clinical history, gross and microscopic lesions. As a result, histology has a pivotal role for the diagnosis process, and not only a “complementary investigation”. Please, delete this expression (complementary investigation) all figures.

Response 9: Done

Point 10: In page 11, second paragraph of section 3.2.1.3. Diagnostic tools and criteria, all Clostridium perfringens types produces a double zone of hemolysis, not only C. perfringens type C. Modify the sentence to make it clear.

Response 10: Done

Point 11: In page 12, in the first line of the topic 3.2.1.1. Aetiology and clinical presentation, please, substitute the word “flora” by “microbiota”.

Response 11: Done

Point 12: In figure 12 (former scheme 6), make clear that the detection of CPB2 does not contribute at all for the diagnosis of C. perfringens type A, as demonstrated in many published studies. The exclusion of all other possible causes and the visualization of a myriad of bacilli in close contact to the enterocytes in the small intestine are the only indication of “POSSIBLE” indication of the C. perfringens type A involvement in the diarrhea.

Response 12: The figure 12 (former scheme 6), was modified as addressed by the Reviewer

Point 13: In page 15, in the first line of the topic 3.3.1. Aetiology and clinical presentation, please, substitute the word “flora” by “microbiota”.

Response 13: Done

Point 14: In page 16, in the first line of section 3.4.1. Aetiology and clinical presentation, delete the word “Neonatal” as coccidiosis can only manifest after 5 days of age. It might give the false impression that it would cause diarrhea in very young piglets. Start the sentence with “Coccidiosis caused….”

Response 14: Done

Point 15: In figure 16 (former scheme 9), if you have histologic lesions indicative of viral infection and PCR positive for Rotavirus, this is confirmatory diagnostic for Rota and does not require any other direct detection of the virus by IHC or ISH. Sometimes these two last tests can be negative and the animal still Rotavirus positive due to exfoliation of enterocytes and reduction of infected cells. Modify the figure accordantly.

Response 15: The figure 16 (former scheme 9), was modified as addressed by the Reviewer

Point 16: Figure 25 (former scheme 12):  PHE is not associated with “fever”, please, delete this word from the figure.

Response 16: The figure 25 (former scheme 12), was modified as addressed by the Reviewer

Point 17: In figure (page) 30, the item (5) of the list of Early microscopic lesions of SD include [89]: substitute “mucus secretion” by “goblet cells hyperplasia”.

Response 17: Done

Point 18: In page 31, in the second paragraph, I believe MALDI-TOF MS does not allow adequate  discriminatory differences among different Brachyspira spp. Please, confirm this information.

Response 18: I confirm that it was reported as MALDI-TOF MS correctly assigned the isolates tested to the genus Brachyspira, identifying the strains as B. hyodysenteriae, B. pilosicoli, B. intermedia and B. innocens (S. Prohaska et al, 2013 - MALDI-TOF MS for identification of porcine Brachyspira species). Added this citation.

Reviewer 2 Report

The manuscript by Luppi et al. describes the etiology and diagnostic approaches of enteric disorders in pigs. The paper is well written. The data are very informative. I think this is a very well-organized paper that would be very attractive to the readers of this journal.

I have only one suggestion: although the schemes are informative, it would be interesting to include a final scheme/table including all pathogens in order to compare the diagnostic tools more appropriate for each pathogen as well as pathognomonic findings (if exist) that may help clinicians or pathologists in the diagnosis.

Minor:

Simple summary: “ The solution of an emteric disease” change to “ The solution to an enteric disease”

Page 11. “This autolysis can destroy the architecture of the tissue and antigens pre-sent in it, thus immunoistochemistry to detect C. perfringens beta-toxin is not the method of choice for routine testing”. Correct the word “immunoistochemistry”

Scheme 5. C.perfringes, include space

Scheme 7 and legend: C.difficile, space

Page 19: “syndrome coronavirus (SADS-CoV) are recognized as causes of gastrointestinal disease in pigs [55].”. There is an extra space

Page 21 “3.6.3. Diagnostic tools and criteria

The main methods for laboratory diagnosis of PECs include polymerase chain reac-tion (PCR) immunofuorescence” Correct “immunofuorescence”

Scheme 10: space missing in “S Typhimurium”

Scheme 12. Title in orange: revise

Author Response

Parma, 6st of January 2022

To the Editor of Animals,

Subject: Response to Reviewer’s comments on the manuscript – Diagnostic approach to enteric disorders in pigs - Manuscript ID: Animals-2117665. Type: Review

Authors: Andrea Luppi, Giulia D’Annunzio, Camilla Torreggiani, Paolo Martelli

Section: Pigs - Special Issue: Gastrointestinal Tract Health in Pigs

Dear Editor,

Thank you for assessing the manuscript “Diagnostic approach to enteric disorders in pigs" and for your positive response on its potential publication in Animals following the proposed revisions suggested by the reviewers. Thank you for the comments and review proposals that led to an improvement of the manuscript.

You will find below the responses to the Reviewer’s comments and specific changes implemented in the manuscript as a result of these comments. All changes made have been highlighted with 'tracked changes'.

I trust that the reviewed version of the manuscript, attached to this letter, will satisfy these comments and is now acceptable for publication in Animals.

Looking forward to hearing from you.

Yours faithfully,

Andrea Luppi

Corresponding author for manuscript submission (Animals-2117665).

Response to Reviewer 2 Comments

The manuscript by Luppi et al. describes the etiology and diagnostic approaches of enteric disorders in pigs. The paper is well written. The data are very informative. I think this is a very well-organized paper that would be very attractive to the readers of this journal.

Point 1: I have only one suggestion: although the schemes are informative, it would be interesting to include a final scheme/table including all pathogens in order to compare the diagnostic tools more appropriate for each pathogen as well as pathognomonic findings (if exist) that may help clinicians or pathologists in the diagnosis.

 Response 1: Done. Table 3, summarizing clinical/pathological findings and diagnostic tools has been added.

Minor:

Point 2: Simple summary: “ The solution of an emteric disease” change to “ The solution to an enteric disease”

Response 2: Done

Point 3: Page 11. “This autolysis can destroy the architecture of the tissue and antigens pre-sent in it, thus immunoistochemistry to detect C. perfringens beta-toxin is not the method of choice for routine testing”. Correct the word “immunoistochemistry”

Response 3: Done 

Point 4: Scheme 5. C.perfringes, include space

Response 4: Done 

Point 5: Scheme 7 and legend: C.difficile, space

 Response 5: Done

Point 6: Page 19: “syndrome coronavirus (SADS-CoV) are recognized as causes of gastrointestinal disease in pigs [55].”. There is an extra space

 Response 6: Done

Point 7: Page 21 “3.6.3. Diagnostic tools and criteria

Response 7: Not clear to me what should be changed

Point 8: The main methods for laboratory diagnosis of PECs include polymerase chain reac-tion (PCR) immunofuorescence” Correct “immunofuorescence”

Response 8: Done
